# Towards a Certificate of Trust: Task-Aware OOD Detection for Scientific AI

**Bogdan Raonic**
Seminar for Applied Mathematics,
ETH Zurich &
ETH AI Center
Zurich, Switzerland

**Siddhartha Mishra**
Seminar for Applied Mathematics,
ETH Zurich &
ETH AI Center
Zurich, Switzerland

**Samuel Lanthaler**

## Abstract

Data-driven models are increasingly adopted in critical scientific fields like weather forecasting and fluid dynamics. These methods can fail on out-of-distribution (OOD) data, but detecting such failures in regression tasks is an open challenge. We propose a new OOD detection method based on estimating joint likelihoods using a score-based diffusion model. This approach considers not just the input but also the regression model's prediction, providing a task-aware reliability score. Across numerous scientific datasets, including PDE datasets, satellite imagery and brain tumor segmentation, we show that this likelihood strongly correlates with prediction error. Our work provides a foundational step towards building a verifiable 'certificate of trust', thereby offering a practical tool for assessing the trustworthiness of AI-based scientific predictions.

## 1 Introduction

Deep learning is rapidly transforming scientific computing. Most problems in this domain involve the prediction of unknown, spatially and/or temporally varying physical properties – such as the temperature distribution in a solid or the flow velocity of a fluid – from given initial or boundary conditions. Traditionally, such problems have been addressed by physical models formulated as partial differential equations (PDEs) Evans (2022), and approximated with bespoke numerical algorithms Quarteroni & Valli (1994). However, data-driven approaches building on neural networks are now increasingly applied to scientific computing Mishra & Townsend (2024), achieving state-of-the-art accuracy in applications like numerical weather forecasting (Bodnar et al., 2025).

This novel data-driven paradigm offers significant advantages, including reduced computational costs and the ability to learn from historical data even when no tractable physical model exists (Lu et al., 2021; Li et al., 2021; Raonic et al., 2023; Pfaff et al., 2021). Nevertheless, purely data-driven approaches also introduce critical drawbacks, primarily concerning prediction reliability. Whereas PDE models reflect fundamental physical laws that remain valid even in extreme, previously unseen conditions, in contrast, data-driven approaches are inherently interpolative, and prediction accuracy can deteriorate for inputs far from the training distribution (Herde et al., 2024).

Machine learning models are typically built on a "closed-world" assumption, expecting test data to share the training data's distribution (in-distribution, or ID). Yet, real-world scientific applications frequently encounter out-of-distribution (OOD) samples that require careful handling (Drummond & Shearer, 2006). As a consequence, deep learning predictions typically lack a *certificate of trustworthiness*, making it difficult to ascertain their accuracy and reliability on real-world inputs.

To address challenges related to ID/OOD distribution shifts, out-of-distribution detection has gained significant attention over the last decade Yang et al. (2024). This has led to the development of a number of OOD detection methods, including classification-, distance-, and density-based approaches. While this topic is extensively studied for tasks such as image classification, its application to regression, which constitute a vast majority of learning tasks in scientific computing, remains severely underexplored.

### 1.1 Contributions

This work addresses the critical need for tools to asses the accuracy and reliability of neural network predictions, particularly for OOD data in scientific and engineering applications. While the ultimate

goal of this research direction is to furnish end-users with reliable "certificates" of prediction quality, the main contribution of the present paper is to propose the following important steps towards this objective,

- We develop and empirically validate a novel approach integrating *any* underlying regression model $\Psi$ with a score-based diffusion model for OOD detection. Our proposed certificate is based on the evaluation of the estimated joint likelihood $p(x, y_{\text{pred}})$, with $y_{\text{pred}}$ being the model's prediction for input $x$.
- Our approach is *zero-shot*, in the sense that it does not require any access to the ground truth samples for the test distribution, for OOD detection. If some ground-truth test samples are available, we can go further than ID vs. OOD detection and provide an *a posteriori* estimate of the underlying prediction error.
- We tailor this method specifically for regression tasks, while also demonstrating its applicability to classification and segmentation problems.
- We perform an extensive evaluation across diverse scientific datasets, including PDE datasets (Wave and Navier-Stokes equations), a humidity forecasting problem utilizing satellite data, image classification benchmarks, and brain tumor segmentation.
- In all cases, we observe a very strong correlation between the model's prediction errors on ID and OOD data, and the estimated joint likelihood $p(x, y_{\text{pred}})$. We also adapt other certificates, derived as aggregated statistics from the probability-flow ODE, to our proposed setting, and show that these resulting baselines also provide satisfactory OOD detection, indicating the efficacy of the proposed approach based on the *joint* input/outputs.

## 2   RELATED WORK

A first approach to OOD detection, with applications to image classification, directly leverages latent features from the trained networks including outputs of the final or earlier layers. For example, Liu et al. (2020); Zhang et al. (2022) define explicit energy scores based on such features. Test samples with lower energy are considered ID and vice versa. A softmax approach for estimating conditional likelihoods is used in (Hendrycks & Gimpel, 2016; Hsu et al., 2020). Other works also use latent features (statistics) to distinguish ID/OOD samples, (Yang et al., 2024) and references therein.

OOD detection can be viewed through epistemic uncertainty, where estimating this uncertainty yields a scalar detection score. Methods like MC-Dropout Gal & Ghahramani (2016) and Rate-In Zeevi et al. (2025) use dropout at train and test time to generate stochastic forward passes that approximate Bayesian inference. Other approaches, such as Chan et al. (2024), use hybrid Bayesian–diffusion methods to estimate epistemic uncertainty.

Density-based methods capture the ID with probabilistic models, flagging inputs from low-density regions as OOD based on likelihoods. Early works employ (mixtures of) Gaussian distributions (Lee et al., 2018; Pleiss et al., 2019). Normalizing flows in classification tasks are leveraged in (Ren et al., 2019; Nalisnick et al., 2019c; Goodier & Campbell, 2023). Some papers estimate likelihoods on latent features with diffusion models Ding et al. (2025); Järve et al. (2025).

Modeling the joint distribution $p(x, y)$ has been explored in Nalisnick et al. (2019a), where a hybrid model coupled a deep invertible transform with a generalized linear model, mainly focusing on OOD detection in classification. A hybrid approach was also put forward by Cao & Zhang (2022). An assessment of likelihood based OOD detection, identifying systematic biases in the context of image classification, is provided in Nalisnick et al. (2019b). Subsequent work revisiting these examples and proposing improvements include Ren et al. (2019); Nalisnick & et al. (2020). Other approaches explore "typicality" Nalisnick et al. (2019c), "local intrinsic dimension" Kamkari et al. (2024), and enhanced normalizing flows via a "approximate mass" penalty Chali et al. (2023) . Beyond likelihood estimation, applications of diffusion models to OOD detection include reconstruction-based approaches (Graham et al., 2023) and work by Heng et al. (2024) (DiffPath), which perform OOD detection based on rate-of-change and curvature of diffusion paths.

The overwhelming proportion of work on OOD detection has been in the vision/image domains with classification as the learning objective. In contrast, there are very few articles that explore how OOD detection (and error certification in general) can be performed in scientific machine learning,

where bulk of the learning tasks are regression-based. A few exceptions to this rule are Elsharkawy & Kahn (2025), who introduce Contrastive Normalizing Flows for parameter estimation for high-energy physics, Fanelli et al. (2022) propose a conditional generative approach for anomaly detection in experimental physics. For drug discovery, Molecular Out-Of-distribution Diffusion (MOOD) Lee et al. (2023) employs a diffusion model to explore chemical space, guiding generation towards novel molecules. Abdi et al. (2025) apply DiffPath to medical image OOD detection.

What this brief literature survey brings out is the scarcity of OOD detection and prediction certification methods for most of scientific machine learning applications. The main goal here is to devise such a method.

## 3  METHODOLOGY

A generic regression task consists in minimizing over parameters $\varphi$, a loss of the form,

$$\mathcal{L} = \int_{\mathcal{X} \times \mathcal{Y}} \ell(y, \Psi_\varphi(x)) \, p(x, y) \, dx \, dy, \tag{1}$$

where $\Psi_\varphi$ is a model of the operator $\Psi$ which defines the ground truth, $\ell$ is the loss function and $p(x, y)$ is the (ground truth) training distribution.

Given an unseen input $x^\star$ with corresponding ground-truth output $y^\star$, our goal is to determine a quantity $c(x^\star)$, to be used as a *certificate*, which correlates with the loss $\ell(y^\star, \Psi_\varphi(x^\star))$. We impose two important requirements: *(i)* $c(x^\star)$ must be computable without knowledge of $y^\star$, and *(ii)* $c(x^\star)$ should indicate to an end user whether they can expect $\ell(y^\star, \Psi_\varphi(x^\star))$ to be small.

**Likelihood as a Certificate.**  To this end of finding a certificate, we provide with a motivating heuristic computation in SI A.1, where under certain assumptions on the training and generalization of the regression model $\Psi_\varphi$ and on the underlying ground truth probability distribution, we derive the following (approximate) relation,

$$\log\left(\ell(y^*, \Psi_\varphi(x^*))\right) \le \alpha \log(\epsilon) - \log(p(x^*, y_{\text{pred}})) + O\left(\epsilon^\beta\right) \tag{2}$$

where $\epsilon > 0$ is the average loss and $y_{\text{pred}} = \Psi_\varphi(x^\star)$, with positive constants $\alpha, \beta$. From the above relation, we immediately observe that i) the error of the prediction $\Psi_\varphi(x^\star)$ nicely relates to (correlates with) the likelihood $p(x^\star, y_{\text{pred}})$ and ii) the error should be small where data are abundant (high likelihood) and can be large where data are scarce (low likelihood).

Moreover, given the *decomposition*, $\log p(x^\star, y_{\text{pred}}) = \log p(x^\star) + \log p(y_{\text{pred}} \,|\, x^\star)$, it follows that the *joint likelihood* as a certificate ensures i) the model $\Psi_\varphi$ should generalize better in regions of high input likelihood $p(x^\star)$ and ii) Task-specific information enters through the conditional likelihood $p(y_{\text{pred}} \,|\, x^\star)$, which captures the intrinsic complexity of predicting $y_{\text{pred}}$ from $x^\star$. The role of each term in this decomposition is explored in SI A.2 for regression tasks for simple one-dimensional functions, where we demonstrate how both terms are essential in designing a good certificate.

Given these heuristic considerations, we will base our certificate on the *joint likelihood* $p(x^\star, y_{\text{pred}})$. However, one immediately runs into the difficulty of determining this joint probability distribution from data. We will approximate this distribution with a diffusion model as described below.

**Diffusion Models.**  Diffusion models map a Gaussian reference distribution to a target distribution $p(z)$. They are commonly implemented using a backward stochastic differential equation (SDE)SDE. However, this SDE also has an equivalent probability flow ODE formulation (Song et al., 2020, Section 4.3):

$$\frac{dz}{dt} = -\frac{1}{2}\sigma_t^2 s(z(t); t). \tag{3}$$

To sample from $p(z)$, we start with samples from a Gaussian prior as initial data and solve the ODE (27). Here $s(z; t) = \approx \nabla_z x \log p_t$ is the so-called *score function* and $\sigma_t$ is the underlying noise level.

As the probability flow ODE (27) transforms a Gaussian prior into the target distribution, it also enables evaluation of the data density $p(z)$. By integrating along the solution path of the ODE, we obtain (Song et al., 2020, Appendix D.2, Eq. (39)):

$$\log p_0(z(0)) = \log p_T(z(T)) - \int_0^T \frac{1}{2}\sigma_t^2 \left(\nabla \cdot s\right)(z(t); t) dt. \tag{4}$$

The divergence term $\nabla \cdot s(z(t); t)$ can be approximated using stochastic estimators, as detailed in (Song et al., 2020, Appendix D.2). In this work, we apply this to the **joint variable** $z = (x, y)$. In practice, the score function is approximated from a trained *denoiser* using Tweedie's formula Karras et al. (2022).

**Computing the Certificate.** As argued above, our certificate is given by the joint likelihood $p(x, y)$. To compute it, we train the denoiser $D_\theta$ of our score-based diffusion model on the available data pairs $(x_n, y_n)$, $n = 1, \ldots, N$. We note that the training of the diffusion model **does not** involve the regression model $\Psi_\varphi$ in any form.

Given any new input $x^\star$, we then first generate the prediction $\Psi_\varphi(x^\star)$ using the regression model, and then we estimate the joint log-likelihood $p(x^\star, y_{\text{pred}})$ by numerically solving the associated probability flow ODE (4), with its score function being estimated by the trained Denoiser $D_\theta$. The certificate computation is also illustrated in Fig. 1 (A,B).

**ID/OOD classification.** While the relation (2) suggests that the test error and the joint likelihood are perfectly correlated, we emphasize that it is a *heuristic* relation and may not hold exactly. Thus, finding an exact formula between the error and the proposed certificate is very difficult. On the other hand, we can still utilize the certificate in the important task of classifying test samples as *in-distribution* (ID) or *out-of-distribution* (OOD), providing the end user with a metric for ascertaining whether the regression model is reliable or not.

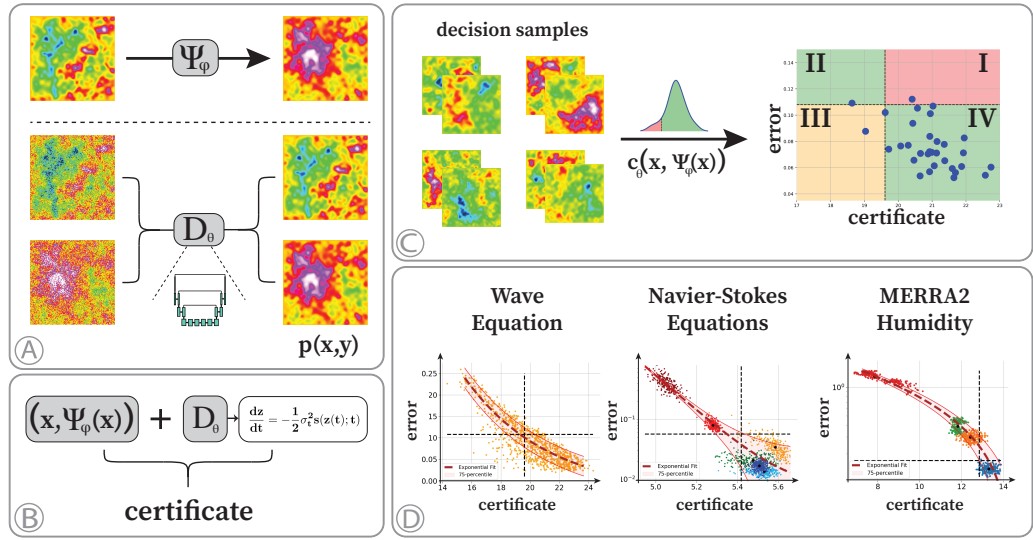

Figure 1: Illustration of the approach: (A) A regression model $\Psi$ and joint diffusion model $D$. (B) Certificate from the probability flow ODE. (C) Classification as ID/OOD based on the certificate (regions II/IV are good). (D) Correlation between error and certificate, with *a posteriori* estimates.

To this end, we first take a small number of *decision samples* from the training distribution and compute the *median* of the corresponding certificate values, denoting it as $l_e$, along with their standard deviation, $\sigma_e$. We define ID samples as those with certificate value greater than $l_e - 1.5\sigma_e$, while OOD samples have values below $l_e - 1.5\sigma_e$. As shown in Figure 1 (C), this procedure defines a vertical boundary between ID/OOD samples, separated according to their certificate values. More formal calibration techniques could also be applied, such as quantile-conformal methods, FPR control, or standard temperature scaling. For *testing purposes*, the horizontal dashed line shows the boundary between small/large errors, here defined as the 95th percentile of errors of the decision samples. Note that the horizontal threshold can be adjusted by the end user, reflecting their chosen tolerance for acceptable error levels. The resulting 4 quadrants in the error vs certificate plane are shown in Figure 1 (C). A good certificate should minimize misclassified samples in regions I and III, corresponding to ID-classified samples with large errors (region I), and OOD-classified samples despite a small prediction error (region III), respectively. This ID/OOD classification procedure provides a quantitative metric to assess the reliability of the certificate. Finally, our proposed overall

algorithm for reliability certification in terms of ID/OOD detection is summarized in Algorithm 1. Further details can be found in SI A.4.

## 4 RESULTS

To assess the proposed approach empirically, we consider a variety of datasets of relevance to scientific computing, including regression on the solution operator for the wave equation, the Navier-Stokes equations and a Humidity Forecast regression dataset, based on real-world data. In addition, we also revisit image classification within the proposed framework, and extend the approach to brain tumor segmentation.

**Wave Equation.** In this experiment, we consider regression on the solution operator of the wave equation with periodic boundary conditions in two spatial dimensions. Initial conditions are obtained from a field with random Fourier coefficients. The distribution is characterized by two parameters $K$ and $r$, where $K$ controls the number of *active* Fourier modes, and $r$ controls the decay rate of Fourier coefficients. Test and training distributions differ in the range of values from which $K$ and $r$ are chosen to generate samples. We refer to SI B.1 for further details on the data distributions.

Once the model $\Psi$ is trained on the training set $X = \{u_{0,n}, \Psi(u_{0,n})\}_{n=1}^{N}$, we then test its performance on the test distribution. For this experiment, the support of the training distribution is a subset of support of the test distribution. Hence, some samples drawn at test-time will be similar to those from training, while others may differ significantly. In addition to the regression model $\Psi_{\varphi}$, we also train a diffusion model $D_{\theta}$ to approximate the joint input/output distribution. The regression model is the CNO architecture of Raonic et al. (2023) and the diffusion model is a UViT type denoiser considered in Molinaro et al. (2025), see SI D for details. The chosen loss function here is the $L_1$-error. Histograms of the estimated likelihood certificate $c_{\theta}(x)$ and $L_1$ errors are illustrated in SI 1 (D).

The approach for ID/OOD detection in the present work hinges on a presumptive correlation between likelihoods and errors: *How does the absolute $L_1$ error correlate with the estimated joint log-likelihood?* We summarize this correlation in Figure 1 (D), where the errors are evaluated for the test distribution. Our results show that samples drawn from the training distribution exhibit higher likelihood values and lower errors compared to those from the test-distribution. Additionally, we observe a very clear correlation between these quantities.

---

**Algorithm 1** OOD Detection with Diffusion Certificates

---

1: Train task model $\Psi_{\varphi}$ on $(x, y)$ and denoiser (diffusion) model $D_{\theta}$ on $p(x, y)$
2: Define certificate $c_{\theta}(x, \Psi_{\varphi}(x))$ via probability-flow ODE (e.g. likelihood as in (4))
3: From training samples, compute (error, $c_{\theta}$) and set ID/OOD boundary
4: **for** test sample $x$ **do**
5: $\quad$ $y_{pred} \leftarrow \Psi_{\varphi}(x)$
6: $\quad$ $c \leftarrow c_{\theta}(x, y_{pred})$
7: $\quad$ **if** $c \geq c_{\text{boundary}}$ **then**
8: $\quad\quad$ classify as ID
9: $\quad$ **else**
10: $\quad\quad$ classify as OOD
11: $\quad$ **end if**
12: **end for**

---

We perform ID/OOD classification as described in Section 3. In addition to the scatter plot of error vs certificate, Figure 1 (D) also shows the resulting classification regions: The vertical dashed line in this plot shows the ID/OOD boundary. Additionally, the horizontal dashed line shows the boundary between small/large errors, which we here define as the 95th percentile of errors in the training distribution. Representative examples of predicted and ground-truth samples from ID and OOD classes can be found in SI B.1, Figures 8 and 9.

For further insight into the results, we split the OOD class into intermediate, or *critical* (CD), where certificates lie in $(l_e - 3\sigma_e, l_e - 1.5\sigma_e)$, and OOD, where certificates fall below $l_e - 3\sigma_e$. To illustrate the ID/CD/OOD separation, we plot joint histograms of $(K, r)$ for ID (left), CD (center), and OOD (right) samples in SI B.1(Figure 11). The ID samples predominantly correspond to high values of the decay parameter $r$, specifically $r \leq 0.75$, which is the minimum observed value of $r$ in the training

set. Critical samples tend to have intermediate values of $r$, whereas OOD samples are characterized by both low $r$ values and typically high $K$ values, regions where the model exhibits the poorest generalization.

*Ablations.* The above results provide strong empirical evidence for the utility of the proposed likelihood certificate on the studied dataset. To better understand the sensitivity of our approach, we performed two ablation studies on the sensitivity to the diffusion model training and number of samples used to determine ID/CD/OOD ranges.

*Sensitivity to the Diffusion Model:* The first ablation examines the extent to which the diffusion model needs to be trained to be effective for OOD detection, with further details in SI B.1.2. The model is trained for $500$ epochs, with final estimated likelihoods shown in Figure 10. We repeat likelihood estimation using intermediate checkpoints with fewer epochs. SI B.1.2 Figure 13 illustrates the progression of the $L_1$ error versus the estimated joint log-likelihood $\log p_\theta(x, y_{\mathrm{pred}})$ throughout training. As the model is trained for more epochs, the estimated likelihood becomes increasingly aligned with the prediction error, with the final model (trained for 500 epochs) showing a pronounced correlation between the two. We also observe that the average estimated log-likelihood over both the training and test distributions increases steadily throughout training, exhibiting a rapid transition during training (cp. SI B.1.2 Figure 14). For the last 100 epochs of training, the model's explanation of the data remains consistent across checkpoints (cp. SI B.1.2, Figure 15). This suggests that once the diffusion model is sufficiently trained, it provides reliable performance for OOD detection.

*Classification Sensitivity:* In previous evaluations, we used 32 samples drawn from the training distribution to classify inputs into ID and OOD categories. We check *what is the number of samples required to achieve reliable classification performance*. This ablation illustrates how the classification boundaries, based on the estimated joint log-likelihood, evolve as the number of randomly selected training samples increases, with results shown in SI B.1.3 Figure 16. With only 4 samples, the classification is conservative, resulting in many test samples being labeled as OOD. As the number of samples used for decision-making increases, the boundaries become progressively more stable and reliable.

*Regression Model Architecture:* In our final ablation, we evaluate the proposed framework using various regression architectures. Instead of the previously used CNO model, we now consider ViT Dosovitskiy et al. (2020), UNet Ronneberger et al. (2015), and C-FNO Molinaro et al. (2025) architectures. The same diffusion model trained in earlier sections is employed throughout. Each regression model is trained on the same dataset used for the CNO experiments (cp. SI B.1.4, Figure 17). In each case, we observe that samples with low likelihoods correspond to high prediction errors, whereas samples with high likelihoods exhibit lower errors across all tested architectures. This indicates that the approach is robust, and does not require a matching regression model architecture and diffusion model backbone.

**Navier-Stokes Equations.** In this experiment, we validate the proposed approach on the time-dependent Navier-Stokes equations with periodic boundary conditions in two dimensions, and with (spectral) viscosity $\nu = 4 \times 10^{-4}$. To this end, we revisit six datasets of varying difficulty, from the papers Raonic et al. (2023) and Herde et al. (2024), termed NS-Sines, NS-Sines Moderate, NS-Shear Layer, NS-Brownian, NS-PwC, with further details provided in SI B.2. For both the regression and diffusion tasks, we employ an *all2all* training strategy, as recommended in Herde et al. (2024).

Labeling of input samples as ID/OOD is performed by the same procedure as in the wave equation. We refer to SI B.2 for additional details related to the time-varying setup of this experiment, and an ablation on autoregressive vs direct formulations. We summarize the correlation between $L_1$-errors and likelihood certificate in Figure 1 (D), where the models are trained on the NS-Mix dataset and tested on a variety of previously unseen datasets.

We again observe a very clear correlation between errors and the likelihood certificate. Additionally, we performed several experiments, where in each experiment we choose a different dataset (or mix of datasets) as our ID training distribution, and we test OOD detection on the other datasets, with results shown in B.2,

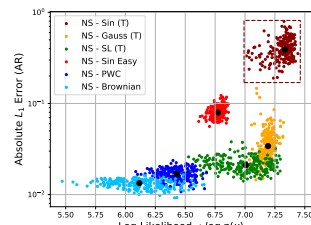

Figure 2: Navier-Stokes. $L_1$ Error vs input-only likelihood $\log p_\theta(x)$ for NS-Mix.

Fig. 19. These results demonstrate that ID/OOD detection works robustly across this range of datasets.

*Insufficiency of $p(x)$ as a certificate.* So far, we have restricted attention to the joint likelihood $p(x, y)$. We here investigate the suitability of $p(x)$ as an alternative certificate. A potential issue with this approach is that the distribution $p(x)$ is completely **task-agnostic**. The task itself could be to solve a PDE, given the input $x$, but it could be something completely different. Therefore the intrinsic difficulty of the task is not incorporated into the input distribution. Moreover, the way we evaluate the trained model is also not incorporated into the certificate. Therefore we do not recommend this approach, given the supporting evidence below.

We analyze the certificate $\log p_\theta(x)$ for the *NS-MIX* problem. In Fig. 2, we present the $L_1$ errors plotted against the estimated log-likelihood. While we observed a clear correlation with errors when using the joint likelihood $p(x, y)$ as a certificate (cp. Fig. 1(D)), no such correlation between $p_\theta(x)$ and the $L_1$-errors is observed. Notably, the NS-Sines dataset receives the highest likelihood scores. However, despite these high likelihoods, the downstream task associated with this dataset remains challenging, resulting in large test errors. This indicates that, in this case, $p(x)$ is not a reliable metric for OOD detection. This conclusion is further supported by SI A.2.1, Table 5, which demonstrates that, in fact, *all task-agnostic baselines* fail. This failure occurs for *all* the certificates based on only the input distribution.

**MERRA-2 Humidity Forecast.** In this experiment, we use MERRA-2 satellite data to forecast surface-level specific humidity Global Modeling and Assimilation Office (GMAO) (2015). Training is performed on a $128 \times 128$ region over South America, using 4h snapshots, in the period 2016–2021 (SI B.3, Fig. 26). The task is to predict humidity 12h ahead. A time-conditioned regression model is trained to forecast up to 60h (15 steps) into the future, and evaluated on 12h predictions (3 steps). In addition, a diffusion model is trained to estimate the joint likelihood $p(x_{t_1}, x_{t_2})$ of humidity snapshots over the same region. We evaluate humidity prediction for 2023 on four test sets (SI B.3, Fig. 26): (1) South America (training region), (2) Australia–Oceania, (3) Africa, and (4) Asia. Due to differing humidity patterns, generalization degrades outside the training domain: performance is best on South America, moderate on Australia–Oceania, and poor on Africa and Asia.

Figure 1(D) plots $L_1$ errors against the likelihood certificate. We observe that the diffusion model assigns high likelihoods, corresponding to low prediction errors, to samples from South America. Samples from Australia receive slightly lower likelihoods and are mostly identified as OOD. As anticipated, the African and Asian datasets fall entirely within the OOD region. In SI B.3, the predicted humidity fields appear overly smooth, lacking fine-scale structures. This is expected, since the regression task is ill-posed and no auxiliary information (e.g., boundary conditions, wind, temperature, pressure) is provided. For comparison, Fig. 25 shows the error histogram of our 12-hour forecasts against a *persistence* baseline (humidity assumed constant). The model clearly outperforms persistence, with its error distribution shifted to the left.

**Classification.** To complement the regression datasets considered before, we next apply our approach to classification tasks. We start with classic image datasets, CIFAR10 and MNIST. We train a classifier $\Psi_\varphi$ to predict a discrete label $y$ from the image $x$. The classifier is trained using a conventional *softmax*-based loss function, maximizing the log-probability corresponding to the true label $y$. During the training of the diffusion model, we concatenate an additional channel containing the constant value $y$ to the $c$ channels of the image $x$.

The classifier predicts log-probabilities $\log p(y \mid x)$ for *each* class label $y$ (the last layer before the softmax is applied). To include full information about classifier outputs during testing, we do *not* define $y_{\text{pred}}$ as the single class label with highest probability. Instead, to define the channel $y_{\text{pred}}$ that is fed into the diffusion model to compute $\log p(x, y_{\text{pred}})$, we sample the individual pixels of $y_{\text{pred}}$ independently from the set of labels, where each pixel value is chosen with probability $p(y \mid x)$.

Figure 3: CIFAR10 Image Classification. Accuracy vs Likelihood Certificate.

In this way, predictions with low confidence introduce variability into the label channel, effectively "corrupting" those samples. Consequently, samples for which the classifier is confident re-

main mostly unaffected. By incorporating uncertain label values,
we effectively *perturb the one-dimensional manifold* on which the labels reside.

*CIFAR10.* In this experiment, we train both a classifier and a diffusion model using the CIFAR dataset, containing 10 distinct classes. We designate one class as out-of-distribution (OOD), which is underrepresented in the training set. The class chosen as the OOD class is *trucks* (the last class). For each in-distribution class, we select approximately 4.5K training samples, with slight variations in the exact number for each class. For the OOD class, we select only $10\%$ of the samples.

In Figure 3 we show the accuracy of the classifier vs. the likelihood certificate. We observe a linear relation between the accuracy and the predicted likelihoods. As expected, the performance of the classifier was the worst for the OOD class (i.e. the *truck* class). Additionally, the classifier was unable to accurately predict the *cat* class (below $60\%$ accuracy), and the diffusion model accurately assigned low likelihood to this class. Two effects combine to yield this result: (1) The classifier is *rarely overconfident in the wrong label*. (2) Even when the classifier is overconfident in the wrong label, the estimated likelihood is still much lower than the ones obtained when the classifier is overconfident in the correct label.

*MNIST.* We repeat this experiment for MNIST. The OOD class is the *number 9*. The results, shown in SI B.7, Figure 32 are similar to the ones obtained in the case of CIFAR10 dataset. Note that the classification task is very easy, so almost all the ID samples are properly classified. Finally, we perform an extensive ablation of our approach on the well-known issues of ID/OOD misclassifications for CIFAR/SVHN identified in Ren et al. (2019), with details in SI B.7.1.

**Segmentation.** In this section, we evaluate our approach on **binary segmentation** tasks (i.e. pixel-wise classification). Our method follows a similar strategy as for classification, with one key distinction: we explicitly reduce the influence of non-semantic pixels by corrupting them with white noise during training. The method is explained in full detail in SI B.8.

Our objective is to perform **brain tumor segmentation** on the **BraTS2020** dataset Menze et al. (2014). This dataset contains 3D brain MRI volumes. The data is divided into two categories: (1) High-grade gliomas (HGG), (2) Low-grade gliomas (LGG). Each brain scan is accompanied by a simplified segmentation mask defined as 0: non-tumor tissue pixels and 1: tumor tissue pixels. We train our segmentation model using brain scans with HGG tumors, from which we select 190 for training, 10 for validation, and 10 for testing. During training, we apply a range of augmentation techniques. We refer to SI B.8 for further details on the datasets and employed augmentation techniques; we also include an ablation on the noise corruption technique.

Our evaluation is conducted on 10 held-out HGG brains and an additional set of 10 LGG brains. For the HGG cases, we evaluate the model not only on FLAIR MRI scans, which were used during training, but also on $T_2$-weighted scans, representing a different MRI modality. For the LGG cases, performance is assessed on both axial (z-axis) slices, aligned with the training direction, and x-axis slices, offering a side view of the brain and allowing us to test the model's generalization to previously unseen anatomical orientations.

SI B.8, Fig. 37 shows the relation between relative $L_1$ segmentation error and our likelihood certificate across four test scenarios. Most low-error cases are correctly classified as ID, while nearly all high-error cases (relative $L_1 \geq 1.0$) are identified as OOD. Furthermore, it is crucial to highlight that our approach effectively identifies OOD samples originating from a **different MRI modality**, namely $T_2$ MRI scans (see subfigure 3 in Figure 37).

Aggregating all datasets, the 2d histogram of error vs. likelihood (SI B.8, Fig. 38(left)) shows high density around low likelihood and errors near $1.0$, i.e., OOD. Low-error points cluster near the threshold but remain ID. The log-likelihood histogram (middle) is right-skewed, favoring higher values. Finally, error histograms (right) confirm that ID samples are mostly low-error, while OOD samples are dominated by high-error cases, with some low-error outliers.

**Quantitative performance metrics and Baselines.** As illustrated in Fig. 1(C), the ID/OOD boundary (vertical) and error boundary (horizontal) divide the error-vs-likelihood scatter plot into 4 quadrants. We consider the *null-hypothesis* that testing samples are OOD and, based on this sub-division, identify true positives (classified OOD, large error), false positives (classified OOD, small error), true negatives (classified ID, small error) and false positives (classified ID, large error). Further details can be found in SI A.4.

| – | – | JLBC | JDPath | JSFNS | JSBDDM | JMSSM | OODC |
|---|---|---|---|---|---|---|---|
| **Wave** | ACC | 0.855 | 0.864 | 0.862 | 0.865 | **0.892** | 0.545 |
| | FPR | **0.040** | 0.108 | 0.095 | 0.108 | 0.066 | 0.395 |
| | FDR | **0.126** | 0.359 | 0.314 | 0.359 | 0.220 | 0.453 |
| | AUROC | 0.936 | 0.912 | 0.916 | 0.913 | 0.946 | – |
| **NS-PwC** | ACC | **0.994** | 0.988 | 0.989 | 0.988 | 0.989 | 0.603 |
| | FPR | **0.001** | 0.002 | 0.002 | 0.002 | 0.002 | 0.142 |
| | FDR | **0.002** | 0.003 | 0.003 | 0.003 | 0.003 | 0.673 |
| | AUROC | **0.999** | **0.999** | **0.999** | **0.999** | **0.999** | – |
| **NS-MIX** | ACC | **0.947** | 0.788 | 0.786 | 0.788 | 0.788 | 0.424 |
| | FPR | **0.009** | 0.022 | 0.020 | 0.021 | 0.020 | 0.090 |
| | FDR | **0.024** | 0.062 | 0.058 | 0.060 | 0.058 | 0.350 |
| | AUROC | **0.992** | 0.918 | 0.886 | 0.913 | 0.891 | – |
| **MERRA2** | ACC | 0.956 | **0.989** | 0.922 | 0.981 | 0.987 | 0.741 |
| | FPR | 0.034 | 0.004 | 0.067 | **0.001** | 0.002 | 0.259 |
| | FDR | 0.046 | 0.006 | 0.086 | **0.002** | 0.003 | 0.518 |
| | AUROC | 0.992 | **0.998** | 0.989 | 0.997 | **0.998** | – |
| **Brain** | ACC | 0.743 | **0.789** | 0.727 | 0.785 | 0.772 | 0.709 |
| | FPR | **0.077** | 0.087 | 0.169 | 0.097 | 0.123 | 0.291 |
| | FDR | **0.253** | 0.297 | 0.580 | 0.332 | 0.422 | 0.291 |
| | ARCB | 0.743 | **0.765** | 0.381 | 0.726 | 0.611 | 0.705 |
| | AUROC | **0.808** | **0.808** | 0.742 | 0.802 | 0.782 | – |
| **Average** | ACC | **0.899** | 0.884 | 0.857 | 0.881 | 0.886 | 0.617 |
| | FPR | **0.033** | 0.045 | 0.071 | 0.046 | 0.043 | 0.224 |
| | FDR | **0.091** | 0.145 | 0.208 | 0.151 | 0.141 | 0.457 |
| | AUROC | **0.945** | 0.927 | 0.906 | 0.925 | 0.923 | – |

Table 1: Performance metrics on scientific datasets for proposed likelihood certificate, and several OOD detection baselines (using joint input/output distribution).

To quantify the performance of the proposed certificate across our experiments, we finally report relevant statistical metrics in Table 1. Specifically, we report the accuracy (measuring correctly classified samples), false positive rate (FPR), and false discovery rate (FDR). To ensure statistical significance of our results, we also report the AUROC metric. The AUROC represents the probability that a randomly selected positive sample receives a higher classifier score than a randomly selected negative one, and is inherently threshold-independent. The proposed likelihood certificate (termed as *JLBC*) is compared to a number of diffusion-based baselines: a curvature-based certificate *JDPath* (see Heng et al. (2024)), a certificate incorporating contributions both from the curvature of the score function and from the score function itself (termed *JSBDDM*) Abdi et al. (2025), the sum of the score functions *Joint Score Function Norm Score* (*JSFNS*, introduced by us in this paper), and a certificate based on sums of norms of the score, referred to in our framework as *JMSSM* Mahmood et al. (2020). All these baselines are still computed by using the denoiser to calculate the score function for the *joint distribution*. All the previous works primarily relied on input-distribution-based approaches. As part of our contribution, we extend these methods to the joint-distribution setting (denoted with *J*) by adapting their approaches accordingly, ensuring a fair and consistent comparison within our framework. We additionally include a non-diffusion baseline *OODC* (see D.1), which requires access to the ground-truth for some *test samples*. For the Wave Equation experiment, we additionally compared our method against two Bayesian-style approaches where the predicted epistemic uncertainty is used for OOD detection, namely MC-Dropout Gal & Ghahramani (2016) and Rate-In Zeevi et al. (2025), both of which use dropout during training and inference to enable stochastic forward passes (i.e., approximate Bayesian inference). Consult SI C for further details on all the baselines.

Throughout all experiments, we find that certificates derived from diffusion models trained on the *joint input/output distribution* robustly classify inputs with large errors to be OOD, as indicated by the low FPR. The results furthermore indicate that the likelihood based certificate is the most robust among these certificates, as demonstrated by it being the best performing approach on average (Tab.

1). Moreover, the JLBC certificate is significantly more accurate for all metrics on the most challenging NS-MIX dataset, where the underlying training and test distributions are both mixtures of multiple distributions. For this task, we also compute the ROC curve and compare it against several baselines (see Figure 4, right). The JLBC demonstrates near-perfect OOD discrimination, whereas the other models show considerably lower ability to distinguish between ID and OOD samples. Regarding the comparison against Bayesian approaches, JLBC clearly surpasses both MC-Dropout and Rate-In in the Wave Equation experiment, delivering substantially higher accuracy and AUROC while also being considerably faster to evaluate (see SI Table 4 and SI B.1.7). Finally, we conduct an ablation study in SI B.1.6 demonstrating that JLBC delivers reliable and stable OOD certificates while requiring only a fraction of a second per sample for certificate computation, enabling fast and robust inference in practice. These findings highlight the utility and potential of the proposed joint input/output approach for identifying problematic predictions across a variety of datasets. Further discussion and ablations on the choice of boundaries can be found in SI A.4 (cp. Table 2).

**A Posteriori estimates on the prediction error.** We reiterate that our proposed approach is *zero-shot* as no access to *any* ground truth test samples is necessary. A natural question that arises is: can we say more in case we have access to the ground truth for some test samples. Revisiting Eqn. (2), we see that the error-(log-)likelihood relation is heuristically an approximate exponential. Hence, we aim to *fit* a scaled and shifted exponential to the error log-likelihood relation for a small number ($\sim$64) of samples of the test distribution for our regression tasks (Wave, NS-Mix and MERRA-2, see SI B.4 for details). We observe from Fig. 1 (D) and SI Fig. 27, that this exponential fit provides a reliable estimate of the error from the likelihood, yielding a quantitative a posteriori error estimate, which can be very useful in scientific applications.

**Inference on Training Distribution.** . In some cases, the objective is to assess the model's generalization ability within its own training distribution. The main challenge here is to identify the *most challenging* samples that still belong to that distribution. In this regard, we perform a posteriori error estimation for the Wave-Eq and NS-PwC experiments using 64 training samples to determine likelihood and error bounds, and a respective relationship between them. Uncertainty bounds of the established relationship are derived via the 75th-percentile rule. For the NS-PwC experiment, we present the error fits in Figure 4. We also examine how the uncertainty bounds depend on the chosen confidence threshold by varying the percentile used to define the bands. As shown in SI Figure 29, increasing the threshold from the 65th to the 95th percentile expands the bounds, capturing more samples but also amplifying the associated uncertainty. For further details, see SI B.4.1.

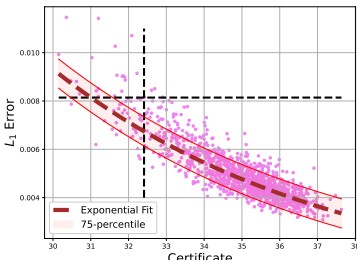 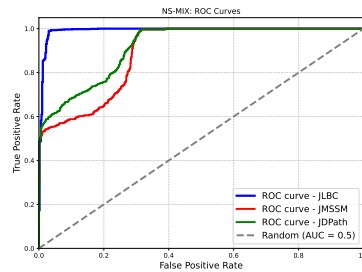

Figure 4: Left: Error fits and corresponding error–certificate plots for the training distributions (NS-PwC experiment). Right: ROC curves for the NS-MIX experiment, where JLBC shows near-perfect OOD discrimination, while other models perform notably weaker.

## 5 CONCLUSION

In this work, we addressed the critical challenge of assessing the reliability of data-driven models in scientific AI, where out-of-distribution failures can have significant consequences. We proposed a novel, task-aware OOD detection method tailored for regression tasks. Our approach leverages a score-based diffusion model to estimate a variety of certificates on the *joint input/output distribution*. This is found to be crucial for an informative reliability score for regression tasks, where methods based on the input distribution $p(x)$ can completely fail. Thus, this work represents a foundational step towards building verifiable "certificates of trust" for AI-based scientific predictions.

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

# A  THEORY AND MOTIVATION

## A.1  MOTIVATION FOR JOINT LOG-LIKELIHOODS AS CERTIFICATES

We are in the setting of (1) and assume that the loss function $\ell$ is of the form,

$$\ell(y, \Psi(x)) = |y - \Psi(x)|^p, \tag{5}$$

for some $1 \leq p < \infty$. In practice, we set $p = 1$ or $p = 2$.

We further assume that there exists a parameter $\varphi^*$, such that the resulting minimized loss is given by,

$$\int_{\mathcal{X} \times \mathcal{Y}} \ell(y, \Psi_{\varphi^*}(x)) \, p(x, y) \, dx \, dy \leq \epsilon << 1. \tag{6}$$

Hence, we assume that the generalization error of the trained model $\Psi^* = \Psi_{\varphi^*}$ is very small.

Next, we fix an $0 < \alpha < 1$ and define the following two sets,

$$A := \{(x, y) \in \mathcal{X} \times \mathcal{Y} : \ell(y, \Psi^*(x))p(x, y) > \epsilon^{\alpha}\}, \quad B := \{(x, y) \in \mathcal{X} \times \mathcal{Y} : \ell(y, \Psi^*(x)) > \epsilon^{\alpha}\}. \tag{7}$$

Clearly $A \subset B$ as $p \leq 1$. Denoting the probability measure $\mathbb{P}$ as,

$$\mathbb{P}(C) = \int_{\mathcal{X} \times \mathcal{Y}} \chi_C(x, y)p(x, y)dxdy, \quad \forall \text{ measurable } C \subset \mathcal{X} \times \mathcal{Y},$$

we have that $\mathbb{P}(A) \leq \mathbb{P}(B)$.

By Chebychev's inequality, we obtain that,

$$\mathbb{P}(A) \leq \mathbb{P}(B) \leq \frac{1}{\epsilon^{\alpha}} \int_{\mathcal{X} \times \mathcal{Y}} \ell(y, \Psi^*(x))p(x, y)dxdy \leq \epsilon^{1-\alpha}. \tag{8}$$

Hence, we also obtain that,

$$\mathbb{P}(A^c) \geq 1 - \epsilon^{1-\alpha} \approx 1. \tag{9}$$

Thus, under the assumption of a well-trained and generalizable model $\Psi^*$, we have, *with very high probability* of $1 - \epsilon^{1-\alpha}$, the event that

$$A^c := \{(x, y) \in \mathcal{X} \times \mathcal{Y} : \ell(y, \Psi^*(x))p(x, y) \leq \epsilon^{\alpha}\}, \tag{10}$$

occurs.

Under the assumption that $p(x, y) \neq 0$, for any $(x, y) \in \mathcal{X} \times \mathcal{Y}$, we can divide in (10) to conclude that, with very high probability, we have a pointwise estimate of the form,

$$\ell(y, \Psi^*(x)) \leq \frac{\epsilon^{\alpha}}{p(x, y)}. \tag{11}$$

Taking logarithms in (11) and observing that both its sides are positive results in the following pointwise estimate (which holds with high probability),

$$\log(\ell(y, \Psi^*(x))) \leq \alpha \log(\epsilon) - \log(p(x, y)), \tag{12}$$

for all $(x, y) \in A^c$

Under the assumption that $\log(p(x, y))$ is locally Lipschitz in $y$, one can expand it around $\Psi^*(x)$ to obtain

$$\log(p(x, \Psi^*(x))) \leq \log p(x, y) + L|y - \Psi^*(x)|,$$
$$\leq \log p(x, y) + L\ell(y, \Psi^*(x))^{\frac{1}{p}}, \quad \text{from(5)} \tag{13}$$
$$\log p(x, y) + O(\epsilon^{\frac{\alpha}{p}}),$$

where the last inequality follows from the fact that $(x, y) \in B^c$.

Plugging (13) into 12, we obtain with high probability that,

$$\log(\ell(y, \Psi^*(x))) \leq \alpha \log(\epsilon) - \log(p(x, \Psi^*(x))) + O\left(\epsilon^{\frac{\alpha}{p}}\right), \tag{14}$$

which is precisely Eqn. (2) of the main text.

Note that the above form Eqn. (14) clearly demonstrates that the loss is controlled in terms of the joint likelihood-based certificate with very high-probability and motivates our use of these certificates.

In deriving (14), we made some key assumptions, namely, i) that the model $\Psi^*$ has very low generalization errors, i.e., it trains and generalizes well in-distribution ii) we have access to the likelihood (or a good approximation of it) and iii) the ground truth probability density function is non-degenerate and log-Lipschitz. In practice, these assumptions may not hold and we need to empirically verify whether a likelihood-based certificate is a good indicator of the error or not. As demonstrated by the many numerical experiments in the main text, this does appear to hold, in general.

Finally, the inequality in (14) suggests that a high likelihood will result in a low error. This fact is consistent with the observations in Table 1 that the ACC and FPR scores therein are very high and very low, respectively. However, given the inequality in (14), we might expect that a low likelihood might correspond to a low error. Indeed, from Table 1, we see that the FDR is, on average, *three times* higher than the FPR, making it consistent with the nature of the inequality in (14).

### A.2 TOY PROBLEM: ILLUSTRATE CONTRIBUTIONS TO JOINT LIKELIHOOD

In the following, we consider simple toy problems, where a simple multilayer-perceptron (MLP) is trained to regress on functions in 1d. In these examples, $x^\star, y^\star \in \mathbb{R}$ are real-valued, and connected by a noisy relationship $y^\star = f(x^\star)$ with function $f$.

In SI A.2.1, we illustrate the importance of $p(x^\star)$ by regressing on simple $f(x)$, but with an *unbalanced input data distribution* $p(x)$. In SI A.2.2, we illustrate the importance of taking into consideration $p(y_{\text{pred}} \mid x^\star)$ in regression tasks. Here, the input distribution $p(x)$ is balanced by construction, but the dependence of $y^\star$ on $x^\star$ is more complex for positive inputs, $x^\star > 0$, than for negative inputs, $x^\star < 0$.

### A.2.1 IMPORTANCE OF $p(x^\star)$.

In this simple example, we will explore 1d regression using a basic two-layer MLP. Our objective is to approximate a function $f : \mathbb{R} \to \mathbb{R}$ from data pairs $(x_i, f(x_i) + \epsilon_i)_{i=1}^N$, where $N$ represents the number of training samples. The noise term $\epsilon_i$ follows a normal distribution $\mathcal{N}(0, 0.1)$, and $x$ is drawn from a specific distribution that we will define shortly.

We are interested in a scenario where the distribution of training inputs exhibits two modes: one that is sampled frequently and another that is sampled much less often. Specifically, we want to explore a dataset where there are many samples for positive values of $x$, while negative values of $x$ are significantly underrepresented. Let us define the density of training inputs to be:

$$ p(x) = \frac{1}{C} \begin{cases} \mathcal{N}(x; 1, 0.5), & x > 0, \\ \nu \cdot \mathcal{N}(x; -1, 0.5), & x < 0. \end{cases} $$

Note that there needs to be some normalization constant $C$ so that the integral of $p$ over $\mathbb{R}$ is 1 (there is also some cutoff at $x = 0$). Here, $\nu$ represents the fraction of less represented mode in the data.

Given a function $f$ that we seek to approximate, we construct our training inputs by first selecting the number of positive samples, $N_+$, and drawing them from $\mathcal{N}(1, 0.5)$. Additionally, we include $\nu N_+$ samples drawn from $\mathcal{N}(-1, 0.5)$ in the training set. For evaluation, we generate two test sets, one for positive samples and one for negative samples, each containing 512 points drawn from $\mathcal{N}(1, 0.5)$ and $\mathcal{N}(-1, 0.5)$, respectively.

First, we fix $\nu = 0.1$. We train an MLP, $f_\theta$, to approximate four different functions. Figure 5 presents the target functions, training samples, prediction errors, and overall performance of the trained MLPs. Across all examples, we set $N_+ = 100$ or $N_+ = 200$. Notably, the performance on the $+$ set is consistently 3 to 10 times better in every case. For the exact error, please take a look at the legend of middle figures.

Next, we fix $\nu = 0.1$ and examine how the errors for the $+$ and $-$ sets change as we vary the number of training samples, $N_+$, for all the target functions. For each point on the graphs, **10**

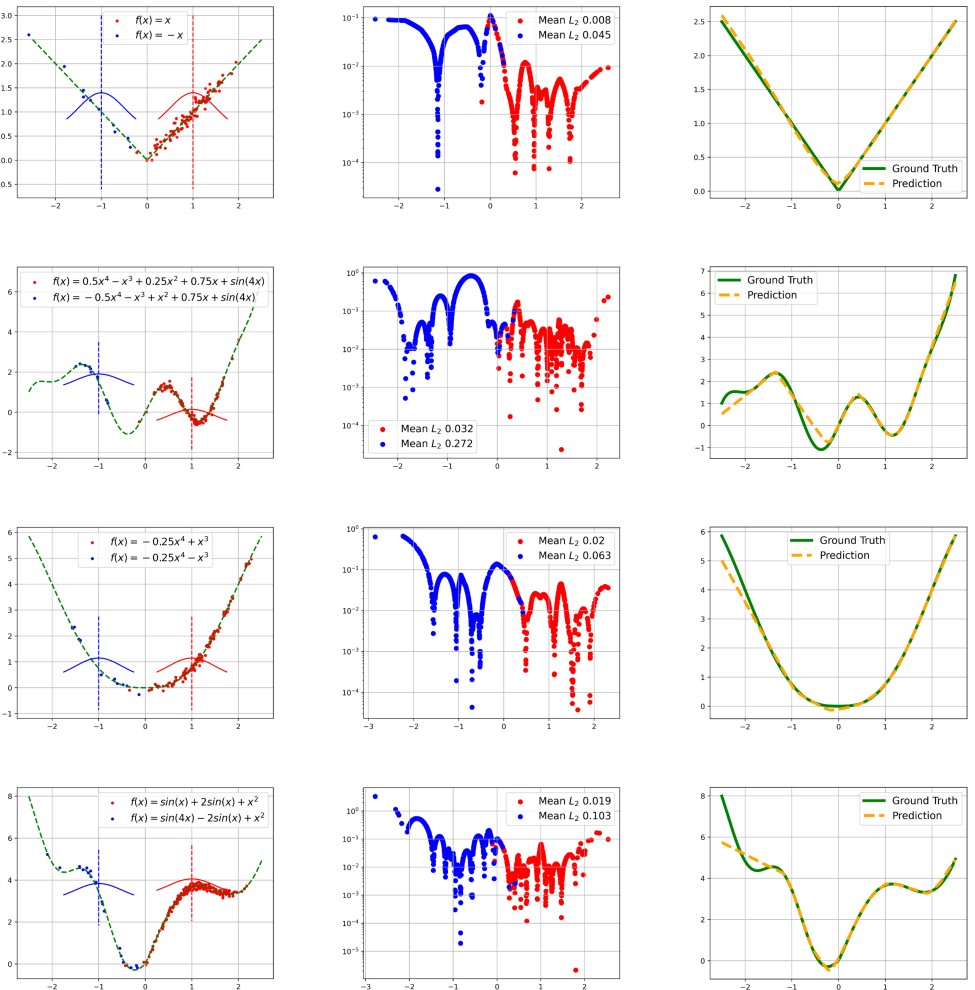

Figure 5: Performance of the trained MLP $f_\theta$ on four different target functions. The figure illustrates the target functions, training samples, prediction errors, and overall model performance. Training is conducted with $N_+ = 100$ or $N_+ = 200$, and the results show that the performance on the $+$ set is consistently 3 to 10 times better. For exact error values, refer to the legend in the middle figures.

**different models are trained**, each time with new training set, the mean $L_2$ error is calculated for each model, and the median of these 10 errors is reported. The results are presented in the left figures of Figure 6. We observe that the $L_2$ errors consistently decrease as $N_+$ increases, which is expected. Similarly, the error for the $-$ class also scales with the number of training samples.

Finally, we set $N_+ = 200$ (or $N_+ = 50$ in case of linear function) and vary the fraction of negative training samples, $\nu$. The right figures in Figure 6 illustrate how the $L_2$ error evolves as $\nu$ increases. We observe that the performance on the $-$ class improves with increasing $\nu$, while the performance on the $+$ class remains largely unaffected. For sufficiently large $\nu$, the errors for both classes become nearly equal. Note that for each point on the graphs, we trained 10 different models and used the same procedure as above to compute the errors.

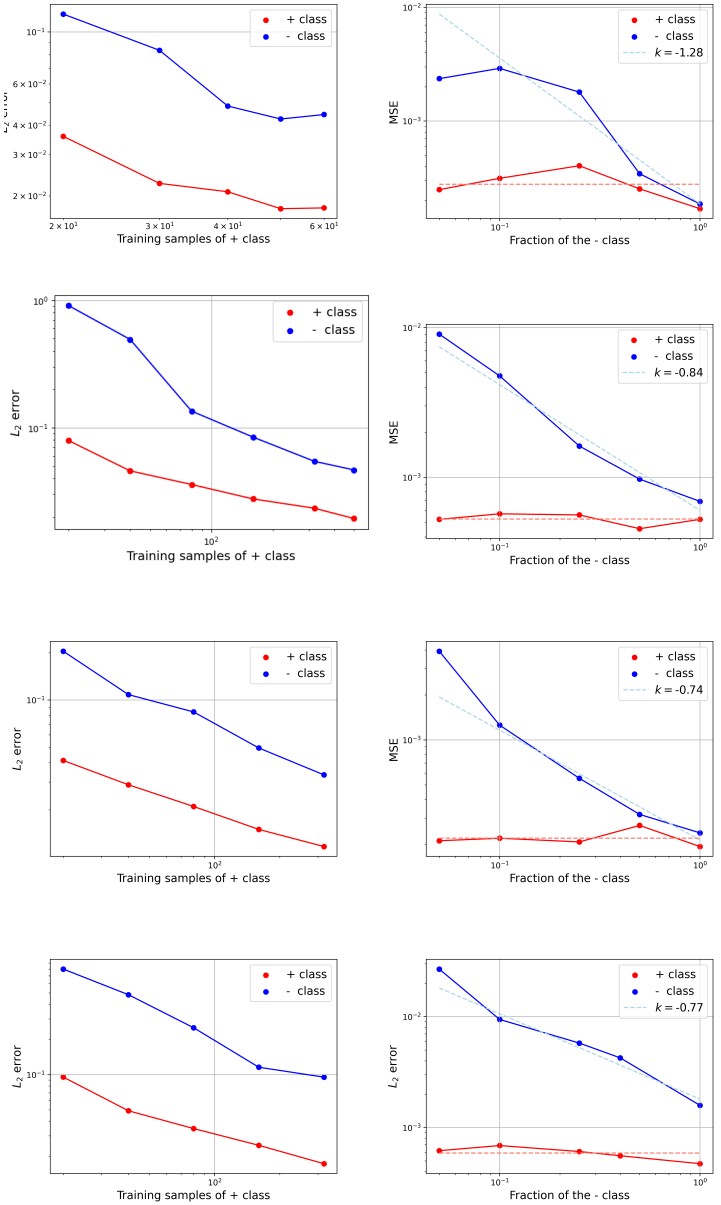

Figure 6: Impact of varying $N_+$ and $\nu$ on $L_2$ errors for the regression problems from Figure 5. For each point on the graphs, **10 different models are trained**, each time with new training set, the mean $L_2$ error is calculated for each model, and the median of these 10 errors is reported. The figures on the left show how errors for the $+$ and $-$ sets change as $N_+$ increases with $\nu = 0.1$, demonstrating a consistent decrease in error. The figures on the right illustrate the effect of increasing $\nu$ while keeping $N_+$ fixed, showing improved performance for the $-$ class while the $+$ class remains mostly unaffected.

(a) Regression model        (b) Diffusion samples (SDE)       (c) log-likelihood $\log p_\theta(x, y)$

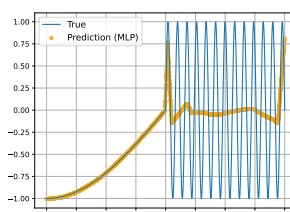 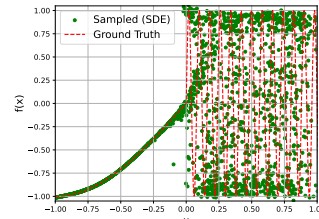 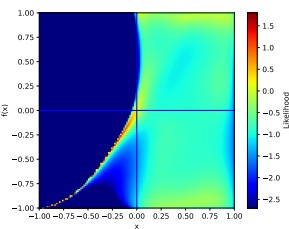

Figure 7: (Left) Ground truth values $f(x)$ and predicted values $f_\theta(x)$, (Center) Samples drawn from the trained diffusion model, (Right) Joint prior log-likelihood $\log p_0(x, y)$.

### A.2.2 IMPORTANCE OF $p(y_{\text{pred}} \mid x^\star)$.

Let us study the function

$$f(x) = \begin{cases} \sin\left(\frac{\pi x}{2}\right), & x < 0 \\ \sin(25\pi x), & x \geq 0 \end{cases} \tag{15}$$

The function $f$ is continuous and exhibits a low-frequency behavior for negative inputs, while it becomes highly oscillatory for positive inputs.

We define the training set as $X = \{(x, f(x))\}_{n=1}^{N}$, where $x \sim \mathcal{U}(-1, 1)$ and $N = 5000$. This means that, on average, half of the dataset represents the low-frequency region of $f$, while the other half corresponds to the high-frequency region. We train a model (an MLP), denoted as $f_\theta$, to approximate the function $f$ using the dataset $X$. The model is trained for 500 epochs. The function $f_\theta$ provides a good approximation of $f$ in the region of negative inputs. However, for positive values of $x$, a phenomenon known as *collapse to the mean value* (as described in Molinaro et al. (2025)) occurs. In this region, where $f$ has a high Lipschitz constant, $f_\theta$ lacks the capacity to accurately approximate the function. The ground truth values of $f$, as well as the predictions of $f_\theta$ are given in the Figure 7 (Left).

Next, we train a score-based diffusion denoiser, $D_\theta$, to generate samples from the *joint distribution* $(x, f(x))$. We expect the diffusion model samples to be concentrated around the curve $(x, f(x))$. For negative values of $x$, this curve occupies a relatively small region of the plane, whereas for positive values of $x$, it spans a much larger portion of the plane. For that reason, for positive values of $x$, we expect the samples to be distributed (almost **uniformly**) in the region $(0, 1) \times (-1, 1)$[1]. In Figure 7, the middle plot displays samples drawn from the trained diffusion model using the probability flow ODE sampler, while the right plot shows samples generated using the Euler-Maruyama SDE sampler. We observe that in the region of negative $x$ values, both techniques yield the samples centered around the graph.

For any point $(x, y)$ in the plane, one can estimate the log-likelihood $\log p(x, y)$ using the instantaneous change of variables formula in the probability flow ODE (see Song et al. (2020)) to get

$$\log p_0(\mathbf{x}(0)) = \log p_T(\mathbf{x}(T)) + \int_0^T \nabla \cdot \tilde{\mathbf{f}}_\theta(\mathbf{x}(t), t) \, dt. \tag{16}$$

In our choice of the forward SDE, we set $T = 1$, while $f_\theta$ can be expressed in terms of the (estimated) score function and the diffusion coefficient. The score function is computed from the denoiser $D_\theta$ using Tweedie's formula. Note that the divergence term inside the integral in 16 can be estimated using Skilling-Hutchinson estimation (see Song et al. (2020) for clarification).

Let us observe a $2d$ uniform, $128^2$ grid in the region $[-1, 1] \times [-1, 1]$. For each grid point, we compute the joint log-likelihood $\log p_0(x, y)$ using the formula 16. The resulting likelihood values are shown in Figure 7 (Right). It is observed that for negative values of $x$, the density is concentrated around the graph, while for positive values of $x$, the probability is distributed across the entire

---

[1]To be more precise, relevant analysis in Molinaro et al. (2025) suggests that the distribution has approximate density $dx \, dy / \sqrt{1 - y^2}$.

region $[0, 1] \times [-1, 1]$, as expected. We note that while estimating the log-likelihoods, an additional correction arises by solving the probability flow ODE backwards in time to obtain log-priors $\log p_T(x_T, y_T)$.

## A.3 LIKELIHOOD ESTIMATION

Joint log-likelihoods (Eq. 16) are computed using the *RK38* solver from the *integrate_torch* library for the initial value problem (and *RK45* in 1d experiments). The divergence term is approximated via a stochastic estimator (see (Song et al., 2020, Appendix D.2)) with *32 Monte Carlo samples*.

## A.4 DECISION BOUNDARIES

After both the problem-specific model and the diffusion model are trained, the likelihood function and error bounds are defined to support decision-making, as well as hypothesis testing. Assume we are given a task-specific model $\mathcal{G} : X \to Y$, a likelihood-estimation function $\mathcal{L}_\theta : (X, Y) \to \mathbb{R}$ (derived from the trained denoiser $D_\theta$), and a **small set** of $M \in \mathbb{N}$ input–output pairs $(x_i, y_i) \in X \times Y$, $i = 1, \dots, M$, sampled from the training distribution.

The decision boundaries illustrated in Figure 1(C) in the main text are derived using a small subset of input–output pairs from the training set (dark blue dots). The **vertical dashed line** represents *certificate threshold*. Samples to the right of this line (higher values) are classified as *in-distribution* (ID), while those to the left (lowe values) are classified as *out-of-distribution* (OOD). The **horizontal dashed line** represents the *error threshold*. Samples with *low certificate values* are expected to lie *above* this line with high probability (they have large prediction errors), while samples with *high certificate values* will lie *below* it, in general (they have small errors). This separation defines four quadrants:

1. **Quadrant I** (upper right, high certificate and high errors): These are the most problematic points. They are classified as ID based on certificate, but their large errors indicate they should be OOD, (i.e. *false positives*).

2. **Quadrant II** (upper left, low certificate + high errors): These are *true positives* for OOD detection, correctly identified as OOD due to low certificate and high error.

3. **Quadrant III** (lower left, low certificate + low errors): These are *false negatives*, samples classified as OOD even though their prediction error is small. These occur as a trade-off to keep Quadrant I small. The horizontal error threshold is chosen not to be too high.

4. **Quadrant IV** (lower right, high certificate + low error): These are *true negatives*. They are correctly identified as ID, with both high certificate and low error.

The objective is to *maximize* the number of true positives (Quadrant II) and true negatives (Quadrant IV) while *minimizing* false positives (Quadrant I). False negatives (Quadrant III) are an acceptable trade-off for (more) strict control over false positives.

There are multiple ways to define the certificate and error boundaries. Given $M$ testing input–output pairs, we first compute the certificate values

$$l_i = \mathcal{L}_\theta(x_i, \mathcal{G}_\varphi(x_i))$$

and the errors

$$e_i = \|y_i - \mathcal{G}_\varphi(x_i)\|_p.$$

We then calculate the median of the certificate values, $m = \mathrm{median}(l_i)$, and their standard deviation, $\sigma = \mathrm{std}(l_i)$. The certificate boundary (vertical line) is defined as

$$l_b = m - \alpha \cdot \sigma,$$

where $\alpha$ is a tunable parameter, set to $\alpha = 1.5$ in all our regression experiments. The error boundary $e_b$ (horizontal line) is defined as the $(100 - \beta)$th percentile of the error values. In our regression experiments, we set $\beta = 0.05$. We also conduct ablation studies to compare alternative methods for defining $l_b$ and $e_b$, and to test the stability of the resulting boundaries across different definitions. Note that the error boundary is introduced only to define the quadrants. One should keep in mind that the error boundary can be defined differently, *depending on the use case and the acceptable margin of error*.

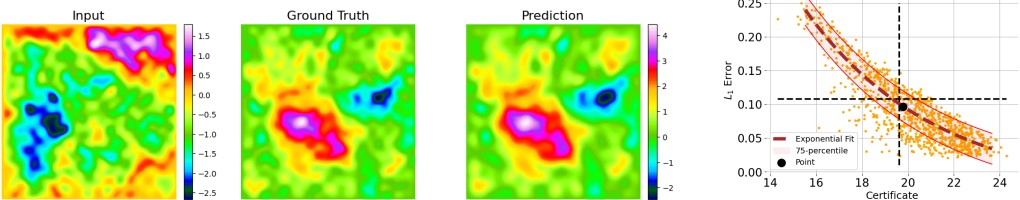

Figure 8: Wave equation. A randomly selected *ID* sample ($q$ dist.). Absolute $L_1$ error is 0.097. The estimated log likelihood is 19.78. Parameters for this samples are $K = 28$ and $r = -0.79$. A posteriori error estimate (defined in B.4) is $0.10 \pm 0.02$.

## B  EXPERIMENTS

### B.1  WAVE EQUATION

**Problem Setup.** In this experiment, we study Wave equation

$$u_{tt} - c^2 \Delta u = 0, \text{ in } D \times (0, T), \quad u_0(x, y) = f(x, y \mid r, K, a_{ij}) \tag{17}$$

with constant speed of propagation $c = 0.1$ and the initial condition given by

$$f(x, y \mid r, K, a_{ij}) = \pi \sum_{i,j=1}^{K} a_{ij} \cdot (i^2 + j^2)^{-r} \sin(\pi i x) \sin(\pi j y), \tag{18}$$

where $K$ is the number of *active* Fourier modes, $r$ is the spectral decay and $a_{ij}$ are coefficients of the respective modes. The exact solution at time $t > 0$ is given by

$$u(x, y, t) = \pi \sum_{i,j}^{K} a_{ij} \cdot (i^2 + j^2)^{-r} \sin(\pi i x) \sin(\pi j y) \cos\left(c \pi t \sqrt{i^2 + j^2}\right), \quad \forall (x, y) \in D.$$

Our objective is to approximate the operator $\mathcal{G} : f \mapsto u(\cdot, T = 5)$.

**Data distributions.** As it is described in the main text, we define the **training distribution** $p$. For each initial condition, the parameters are distributed as $r \sim \mathcal{U}(0.75, 0.85)$, $K \sim \mathcal{U}_{\text{discrete}}(20, 28)$, and $a_{ij} \sim \mathcal{U}(-1.0, 1.0)$. Once the model $\mathcal{G}_\varphi$ is trained on the training set $X = (f_n, \mathcal{G}(f_n))_{n=1}^{N}$, we want to test its performance on the **testing distribution** $q$. For each initial condition, the parameters are distributed as $r \sim \mathcal{U}(.675, 0.925)$, $K \sim \mathcal{U}_{\text{discrete}}(16, 32)$, and $a_{ij} \sim \mathcal{U}(-1.0, 1.0)$. We observe that the $\text{supp}(p) \subset \text{supp}(q)$. Therefore, we expect that some samples drawn from distribution $q$ will be similar to those from $p$, while others may differ significantly. Note that we use only $N = 1000$ samples in the training set.

**Wave Equation - Critical Region (CD)** To better analyze the intermediate region for the Wave equation, we further split the OOD class. We define a *critical* (CD) subset where certificate values fall within $(l_b - 3\sigma, , l - 1.5\sigma)$, while samples with certificates below $l - 3\sigma$ are classified as (pure) OOD.

#### B.1.1  JOINT LOG LIKELIHOOD VS $L_1$ ERROR

We show an example of a predicted and a ground-truth sample from ID and OOD classes in Figures 8 and 9. We observe that the parameter $K$ and the decay $r$ of the ID sample in Figure 8 align with the parameter group of the $p$-distribution. The OOD sample in Figure 9 corresponds to $K = 31$, a value not encountered during training, and is associated with $r = -0.85$ decay. This leads to inaccurate model predictions, as indicated by the error and the low likelihood value.

We show a scatter plot of the estimated likelihood certificate vs the $L_1$ error in Figure 10.

We show the *joint* histograms of the parameter values $(K, r)$ for ID samples (left), critical samples (center), and OOD samples (right) (see Figure 11). The ID samples predominantly correspond to high values of the decay parameter ($r \leq 0.75$) which is the minimum observed value of $r$ in the training set. Critical samples tend to have intermediate values of $r$, whereas OOD samples are described by both low $r$ values and usually high $K$ values.

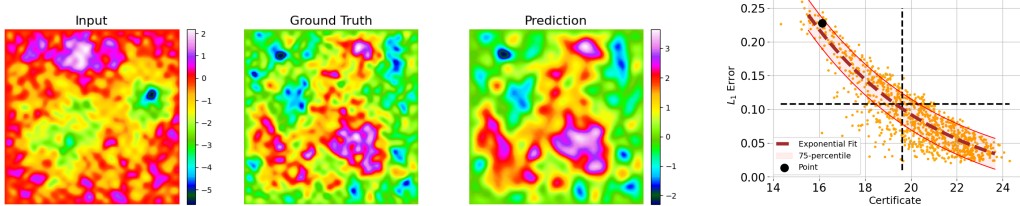

Figure 9: Wave equation. A randomly selected *OOD* sample ($q$ dist.). Absolute $L_1$ error is $0.227$. The estimated log likelihood is $16.12$. Parameters for this samples are $K = 31$ and $r = -0.85$. A posteriori error estimate (defined in B.4) is $0.21 \pm 0.02$.

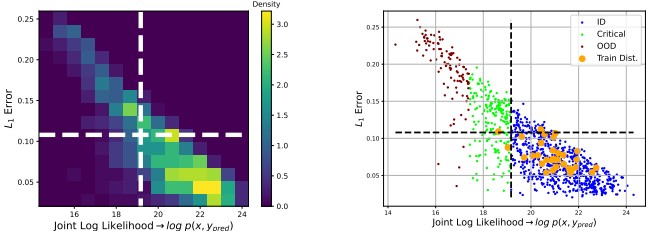

Figure 10: Wave equation. Likelihood–error plane illustrating in-distribution (ID) and out-of-distribution (OOD) classification boundaries, with quadrants indicating true/false positives and negatives.

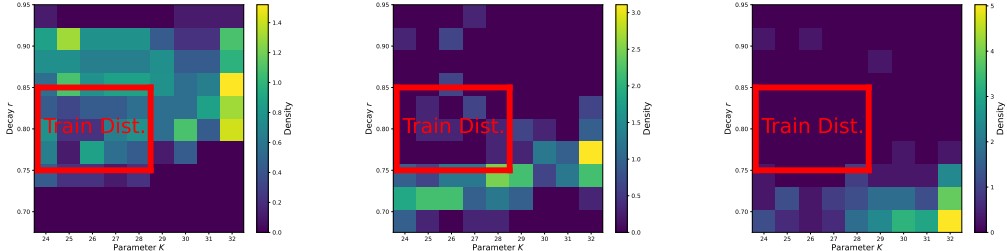

Figure 11: Wave equation. 2d histograms of the values of the parameter $K$ and the decay $r$ for ID samples (left), critical samples (middle) and OOD samples (right). The rectangular region in red represents parameters of the *training distribution*.

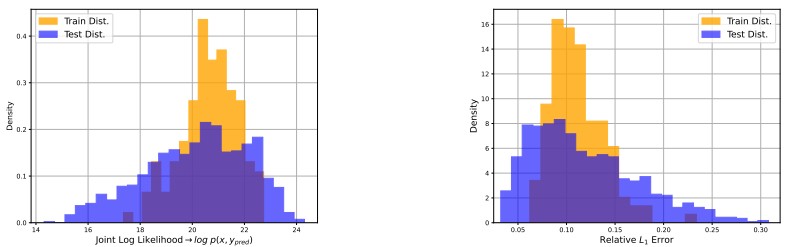

Figure 12: Wave equation. Left: Histogram of estimated likelihoods of the samples drawn from $p$ distribution (training) and $q$ distribution (testing). Right: Histogram of relative $L_1$ errors for the same samples.

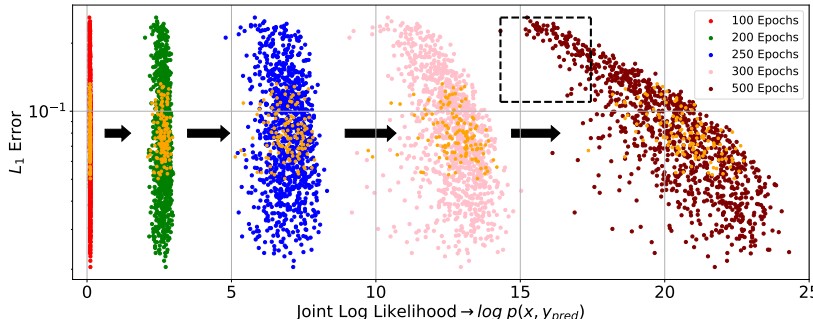

Figure 13: Evolution of the joint log-likelihood $\log p(x, y_{\text{pred}})$ versus the $L_1$ error across training checkpoints of the diffusion model. Likelihoods are estimated using models trained for 100, 200, 250, 300, and 500 epochs. As training progresses, the joint likelihood estimates become more informative for error detection, with the final model (500 epochs) exhibiting a clear correlation between likelihood and prediction error.

### B.1.2 SENSITIVITY TO THE DIFFUSION MODEL

We now examine the extent to which the diffusion model needs to be trained to be effective for OOD detection. We train the model for 500 epochs, corresponding to approximately 8k gradient steps. The estimated likelihoods shown in Figure 10 of the main text are obtained from the diffusion model trained for 500 epochs. We now repeat the likelihood estimation using intermediate checkpoints of the model trained for less epochs. Figure 13 illustrates the progression of the $L_1$ error versus the estimated joint log-likelihood $\log p_\theta(x, y_{\text{pred}})$ throughout training. As the model is trained for more epochs, the estimated likelihood becomes increasingly correlated with the prediction error, with the final model (trained for 500 epochs) showing a noticeable correlation between the two.

We observe that the average estimated log-likelihood over both the training and testing distributions increases steadily throughout training. Figure 14 shows the evolution of the median estimated log-likelihood on the training distribution (red curve) alongside the validation EMA loss (blue curve). In the beginning, the median log-likelihood remains close to zero for the first 150 epochs. It then rises over the next 250 epochs, before gradually saturating in the end of training. The saturation coincides with the plateauing of the validation loss.

In Figure 15, the model evaluated at epoch 400 is shown on the left, and at epoch 500 on the right. The two plots are nearly identical, which indicates that the model's capability to explain the data remains consistent across these two checkpoints. This suggests that once the diffusion model is sufficiently trained, it provides reliable performance for OOD detection.

### B.1.3 CLASSIFICATION SENSITIVITY

In this section, we address the following question: What is the number of samples required to achieve reliable classification performance?

Figure 16 shows how the classification boundaries, based on the estimated joint log-likelihood, evolve as the number of randomly selected training samples increases. With only 4 samples, the classification is conservative, resulting in many test samples being labeled as OOD. As the number of samples used for decision-making increases, the boundaries become more stable and more reliable. At 128 samples, the classification boundaries are well-formed and lead to good performance.

### B.1.4 REGRESSION MODEL ARCHITECTURE ABLATION

In this section, we evaluate the proposed framework using various regression architectures. Instead of the previously used CNO model, we now consider ViT Dosovitskiy et al. (2020), UNet Ronneberger et al. (2015), and C-FNO Molinaro et al. (2025) architectures (see D.4 for architectural details). The same diffusion model trained in earlier sections is used here. Each regression model is trained on the same dataset used for the CNO experiments. Figure 17 presents the $L_1$ errors plot-

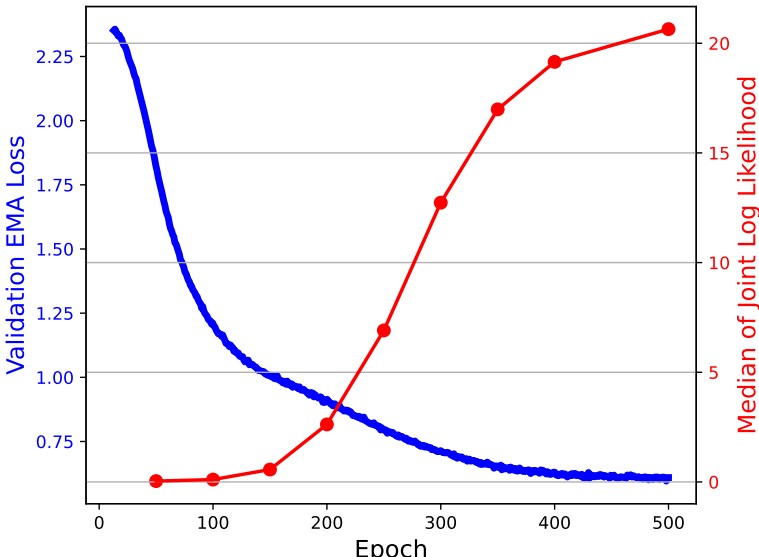

Figure 14: Evolution of the median estimated joint log-likelihood on the training distribution (red) and the EMA validation loss (blue) during the training. The estimated log-likelihood remains low during the initial phase, increases between epochs 150 and 400, and saturates as the validation loss begins to plateau.

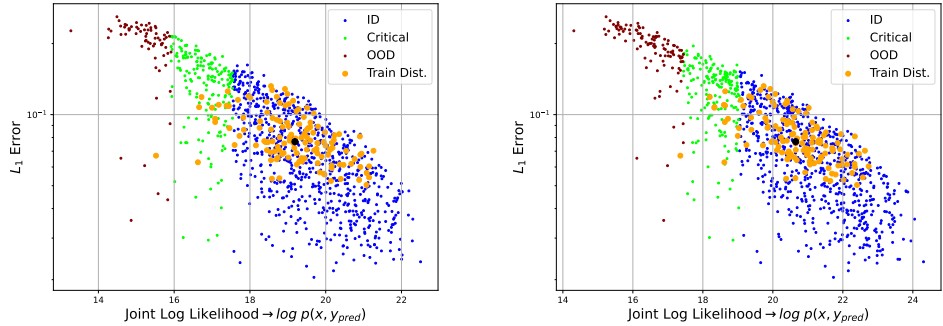

Figure 15: Comparison of the $L_1$ error versus estimated joint log-likelihood $\log p(x, y_{\text{pred}})$ at training epochs 400 (left) and 500 (right). The similarity between the two plots indicates that the model's behavior stabilizes, and the likelihood estimates remain consistent once sufficient training is done.

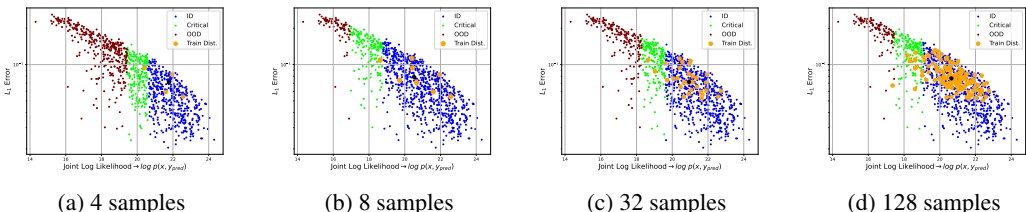

(a) 4 samples       (b) 8 samples       (c) 32 samples       (d) 128 samples

Figure 16: Effect of the number of training samples on the stability of classification boundaries based on the estimated joint log-likelihood $\log p(x, y_{\text{pred}})$. Each subplot shows the $L_1$ error versus estimated log-likelihood for different numbers of randomly selected training samples: 4, 8, 32, and 128.

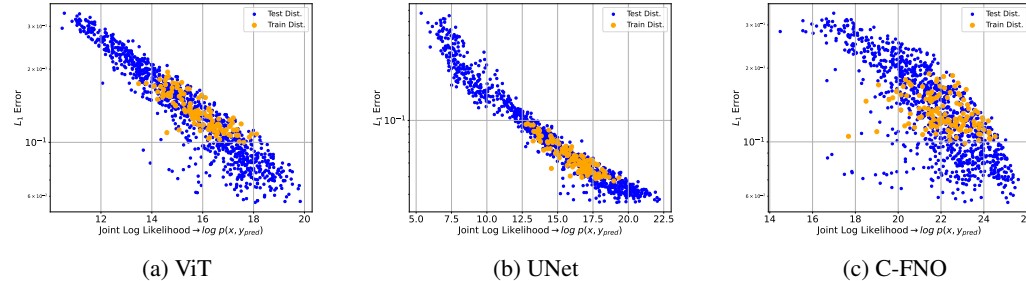

(a) ViT                    (b) UNet                    (c) C-FNO

Figure 17: Comparison of $L_1$ errors and estimated joint log-likelihoods $\log p(x, y_{\text{pred}})$ for different regression architectures (ViT, UNet, C-FNO) using the same diffusion model. While low likelihoods consistently correspond to high-error samples within each model, the absolute likelihood values are not comparable across models.

| $\beta_{ERR}$ | $\beta_L$ | $\alpha_L$ | Accuracy | FPR | FNR |
|---|---|---|---|---|---|
| 0.05 | – | 1.5 | 0.855 | 0.040 | 0.105 |
| 0.01 | – | 1.5 | 0.861 | 0.026 | 0.113 |
| 0.25 | – | 1.5 | 0.813 | 0.133 | 0.054 |
| 0.05 | – | 1.0 | 0.827 | 0.019 | 0.154 |
| 0.01 | – | 1.0 | 0.821 | 0.012 | 0.117 |
| 0.25 | – | 1.0 | 0.819 | 0.095 | 0.086 |
| 0.05 | – | 2.0 | 0.865 | 0.069 | 0.066 |
| 0.01 | – | 2.0 | 0.877 | 0.054 | 0.069 |
| 0.25 | – | 2.0 | 0.777 | 0.186 | 0.037 |
| 0.05 | 0.05 | – | 0.857 | 0.061 | 0.082 |
| 0.05 | 0.01 | – | 0.851 | 0.106 | 0.043 |
| 0.05 | 0.25 | – | 0.770 | 0.010 | 0.220 |

Table 2: Wave equation. Performance metrics (accuracy, FPR, and FNR) for different configurations of the error-boundary percentile $\beta_{\text{ERR}}$ and the likelihood-boundary parameters: either percentile-based ($\beta_L$) or median-and-standard-deviation-based ($\alpha_L$) definitions.

ted against the estimated joint log-likelihoods $\log p_\theta(x, y_{\text{pred}})$ for the different models. Similar to the CNO case, we observe that samples with low likelihoods correspond to high prediction errors, whereas samples with high likelihoods exhibit lower errors across all tested architectures.

We find that this method cannot be reliably used for model selection. Although the estimated log-likelihoods for the C-FNO model are higher than those for the ViT and UNet models, its corresponding prediction errors are also higher. This indicates that only the relative likelihoods within a single model are meaningful, while comparisons of estimated likelihoods between different models are not interpretable.

### B.1.5  DECISION BOUNDARIES ABLATION

We now analyze the dependency and stability of the accuracy, FPR, and FNR as the positions of the likelihood certificate and error boundaries are varied (defined in A.4). While the error boundary is always defined using a percentile-based approach, for the likelihood certificate boundary we compare the *median and std* method with an alternative *percentile-based* definition. Table 2 shows the variation in accuracy, FPR, and FNR as the parameters for the error- and likelihood-boundary estimations are adjusted. We find that accuracy remains relatively stable, even when $\beta_{\text{ERR}}$ is as high as 0.25. The best results are achieved with $\alpha_L = 1.5$ and $\beta_{\text{ERR}} \in \{0.01, 0.05\}$. One should keep in mind that the error boundary can also be specified manually. This section presents an ablation study on our approach to defining this boundary.

### B.1.6   COMPUTATIONAL COMPLEXITY OF THE JLBC FOR OOD DETECTION

As discussed in SMA.3, our certificate estimation procedure uses the RK38 solver. We use a *single* integration step from $t = 0$ to $t = 1$. Because RK38 is a fourth-order Runge–Kutta method, this requires only *four internal substeps* to solve the probability-flow ODE. The Skilling–Hutchinson divergence is estimated using a random tensor of size 32. This choice was made in a (largely) heuristic manner. To evaluate the impact of this choice, we perform an ablation study on the random-tensor size. All inference experiments were conducted on a single RTX 4090 GPU. The performance metrics and samplewise certificate computation times are reported in Table 3.

The results indicate that the complete certification process requires only a fraction of a second per sample. The metrics remain highly stable even when the random-tensor size is reduced to 2, in which case the per-sample inference time is approximately 0.02s. Importantly, the diffusion model requires no retraining or finetuning at inference time, as the proposed method identifies ID/OOD samples in a fully zero-shot manner.

| JLBC Tensor size | Certificate time [s] | AUROC | ACC |
|:---:|:---:|:---:|:---:|
| 32 | 0.211 | 0.936 | 0.855 |
| 8 | 0.062 | 0.936 | 0.859 |
| 2 | 0.020 | 0.937 | 0.857 |

Table 3: Computation time and performance of the JLBC certificate across different random-tensor sizes.

### B.1.7   COMPARISON WITH BAYESIAN APPROACHES

The OOD detection problem can be interpreted through the epistemic uncertainty point of view. High epistemic uncertainty (that indicates the model's lack of knowledge about the system) typically indicates that an input lies outside the training distribution. Estimating this uncertainty and using it as a scalar score enables Bayesian models to act as OOD detectors. In this context, methods such as MC-Dropout Gal & Ghahramani (2016) and Rate-In Zeevi et al. (2025) use dropout during both training and inference, allowing for stochastic forward passes that approximate Bayesian inference by randomly sampling subnetworks.

We test our model on the Wave Equation experiment against MC-Dropout and Rate-In. MC-Dropout estimates uncertainty by performing multiple stochastic forward passes with dropout activated during inference (approximating a Bayesian ensemble). The Rate-In method can be viewed as a more advanced variant of MC-Dropout, where the dropout rates used during inference are adaptively tuned. This adaptation increases inference time compared to standard MC-Dropout, but results in higher accuracy and AUROC. We also re-evaluate the JLBC model (marked with $\star$ in Table 4) using a newly trained version that includes dropout ($p = 0.1$). Overall, the diffusion-based approach remains dominant, achieving much higher performance while requiring only about $0.02$s per sample for certificate computation. This computation time is roughly five times faster than Rate-In despite its lower accuracy. Note that model accuracies are computed using a fixed threshold corresponding to *the mean plus/minus 1.5 standard deviations of the score*, while the AUROC metric remains threshold-independent. Figure 18 shows the histograms of error versus certificate values for JLBC, MC-Dropout, and Rate-In. Among the three, JLBC provides the clearest separation between ID and OOD samples. The Rate-In performs second best and MC-Dropout shows the weakest distinction in this setting.

| JLBC* Tensor size | Certificate time [s] | AUROC | ACC |
|---|---|---|---|
| 32 | 0.211 | 0.955 | 0.873 |
| 2 | 0.020 | 0.955 | 0.869 |
| MC-Dropout Tensor size | Certificate time [s] | AUROC | ACC |
| 32 | 0.028 | 0.526 | 0.407 |
| 2 | 0.002 | 0.642 | 0.676 |
| Rate-In Tensor size | Certificate time [s] | AUROC | ACC |
| 128 | 0.240 | 0.809 | 0.742 |
| 32 | 0.150 | 0.816 | 0.762 |
| 2 | 0.120 | 0.714 | 0.693 |

Table 4: Comparison of diffusion-based certificates, MC-Dropout, and Rate-In approaches across different random tensor sizes. JLBC uses random tensors for estimating the divergence term in the probability-flow ODE, whereas the other two methods use them for Monte Carlo estimation. The diffusion certificates achieve high accuracy and AUROC with low computation time, while MC-Dropout and Rate-In provide weaker uncertainty estimates. Rate-In method offers moderate improvements over MC-Dropout at the cost of higher runtime. With sufficiently large tensor sizes used during Rate-In inference, the performance eventually reaches a saturation point.

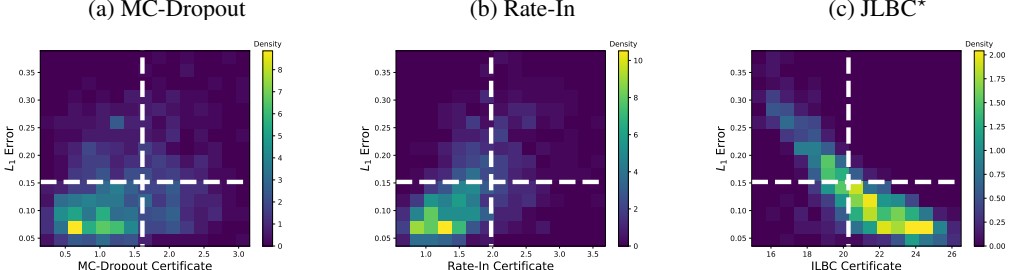

Figure 18: Histogram of error versus certificate values across different methods. JLBC and Rate-In are evaluated using 32 random samples, while MC-Dropout uses 2 samples (corresponding to its best-performing configuration). JLBC exhibits the strongest separation between ID and OOD samples, followed by Rate-In. The MC-Dropout performs the weakest under these settings.

## B.2 NAVIER-STOKES

**Problem Setup.** In this experiment, we study Navier-Stokes equations

$$u_t + (u \cdot \nabla)u + \nabla p = \nu \Delta u, \quad \text{div } u = 0, \tag{19}$$

in the torus $D = \mathbb{T}^2$ with periodic boundary conditions and viscosity $\nu = 4 \times 10^{-4}$, only applied to high-enough Fourier modes (those with amplitude $\geq 12$). The data is taken from the papers Raonic et al. (2023) and Herde et al. (2024).

In this section, we validate our intuition on time-dependent 2D Navier-Stokes equation problems. To achieve this, we define six datasets of varying difficulty (mainly taken from Herde et al. (2024)), namely:

1. **NS-Sines**. We consider the following initial conditions,

$$
\begin{aligned}
u_x^0(x, y) &= \sum_{i,j=1}^{p} \frac{\alpha_{i,j}}{(2\pi(i+j))^q} \sin(2\pi i x + \beta_{i,j}) \sin(2\pi j y + \gamma_{i,j}) \\
u_y^0(x, y) &= \sum_{i,j=1}^{p} \frac{\alpha_{i,j}}{(2\pi(i+j))^q} \cos(2\pi i x + \beta_{i,j}) \cos(2\pi j y + \gamma_{i,j})
\end{aligned}
\tag{20}
$$

   where the random variables are chosen as $\alpha_{i,j} \sim \mathcal{U}_{[-1,1]}$, $\beta_{i,j} \sim \mathcal{U}_{[0,2\pi]}$, and $\gamma_{i,j} \sim \mathcal{U}_{[0,2\pi]}$. The number of modes $p$ is chosen to be $p = 10$, while the spectral decay is $q = 1/2$.

2. **NS-Sines Moderate**. The initial conditions have the same form as in 20, but with the spectral decay $q = 1$. The higher order modes are dampened to a greater extent, making the solution less chaotic.

3. **NS-Gauss**. Given a two-dimensional velocity field $u = (u_x, u_y)$, its *vorticity* is given by the scalar $\omega = curl\ u = \partial_x u_y - \partial_y u_x$. We specify the initial conditions in terms of the vorticity, given by,

$$\omega_0(x, y) = \sum_{i=1}^{p} \frac{\alpha_i}{\sigma_i} \exp\left(-\frac{(x - x_i)^2 + (y - y_i)^2}{2\sigma_i^2}\right) \tag{21}$$

   where we chose $p = 100$ Gaussians with $\alpha_i \sim \mathcal{U}_{[-1,1]}$, $\sigma_i \sim \mathcal{U}_{[0.01,0.1]}$, $x_i \sim \mathcal{U}_{[0,1]}$, and $y_i \sim \mathcal{U}_{[0,1]}$.

4. **NS-Shear Layer**. We take as initial conditions the shear layer,

$$
u_0(x, y) =
\begin{cases}
\tanh\left(2\pi \frac{y - 0.25}{\rho}\right) & \text{for } y + \sigma_\delta(x) \leq \frac{1}{2} \\
\tanh\left(2\pi \frac{0.75 - y}{\rho}\right) & \text{otherwise}
\end{cases}
\tag{22}
$$
$$v_0(x, y) = 0$$

   where $\sigma_\delta : [0, 1] \to \mathbb{R}$ is a perturbation of the initial data given by

$$\sigma_\delta(x) = \xi + \delta \sum_{k=1}^{p} \alpha_k \sin(2\pi k x - \beta_k). \tag{23}$$

   The parameters are chosen to be $p \sim \mathcal{U}_{\{7,8,\dots 12\}}$ $\alpha_k \sim \mathcal{U}_{[0,1]}$, $\beta_k \sim \mathcal{U}_{[0,2\pi]}$, $\delta = 0.025$, $\rho \sim \mathcal{U}_{[0.08,0.12]}$, and $\xi \sim \mathcal{U}_{[-0.0625,0.0625]}$.

5. **NS-Brownian**. We generate Brownian Bridges directly in Fourier space with the following method:

$$W(x) = \sum_{|\mathbf{k}|_\infty \leq N} \frac{1}{\|\mathbf{k}\|_2^{\frac{3}{2}}} \sum_{m,n,\ell \in \{0,1\}} \alpha_k^{(mn\ell)} \text{sc}_m(x)\text{sc}_n(x)\text{sc}_\ell(x) \tag{24}$$

   where

$$
\text{sc}_i(x) =
\begin{cases}
\sin(x) & \text{for } i = 0 \\
\cos(x) & \text{for } i = 1
\end{cases}
\tag{25}
$$

   and the $\alpha_k^{(mn\ell)} \sim \mathcal{U}_{[-1,1]}$. These Brownian Bridges are propagated through the discretized Navier-Stokes system from time $t = -0.5$ to $t = 0$.

| – | – | LBC | DPath | SFNS | SBDDM | MSSM | Joint LBC |
|---|---|---|---|---|---|---|---|
| **NS-MIX** **p(X)** | ACC | 0.404 | 0.487 | 0.486 | 0.487 | 0.484 | 0.947 |
| | FPR | 0.187 | 0.345 | 0.339 | 0.345 | 0.343 | 0.009 |
| | FDR | 0.518 | 0.994 | 0.976 | 0.994 | 0.988 | 0.024 |

Table 5: Approximation of $p(x)$ used for OOD detection for NS-MIX fails completely for diffusion-based baselines. We here include our proposed Joint LBC (JLBC) based on estimating $p(x, y_{\text{pred}})$ as a reference for comparison.

6. **NS-PwC**. The initial vorticity is assumed to be constant along a uniform (square) partition of the underlying domain and is given by,

$$\omega_0(x, y) = c_{i,j} \text{ in } [x_{i-1}, x_i] \times [y_{j-1}, y_j] \tag{26}$$

for $x_i = y_i = \frac{i}{p}$ for $i = 0, 1, 2, ..., p$, and $c_{i,j} \sim \mathcal{U}_{[-1,1]}$. The number of squares in each direction was chosen to be $p = 10$.

Each dataset consists trajectories that are made of 11 solution snapshots (input + 10 solution snapshots). Note that in Herde et al. (2024), the original trajectories have a length of 21, but we subsampled them to a length of 11 by selecting every other snapshot in time. For both the regression and diffusion tasks, we use an *all2all* training strategy, as recommended in the original work.

For the **NS-MIX** dataset, training is conducted on a combination of:

- NS-Sines
- NS-Gauss
- NS-Shear Layer

The model is trained on full trajectories, with 18K trajectories in total, yielding nearly 3M I/O pairs.

For the **NS-PwC** dataset, training uses 5K trajectories of length 8 (the first 8 snapshots). Figure 19 shows the $L^1$ error vs the likelihood certificate for the two experimental settings. Note that the decision boundaries for the NS-MIX dataset are derived only from the NS-Gauss and NS-Shear Layer datasets. Although the model was trained on the NS-Sines distribution as well, its errors remain very large. This is because NS-Sines requires much more training trajectories than 18k to achieve (highly) accurate predictions.

In both NS-MIX and NS-PwC, the models are evaluated across all six distributions described above. The final evaluation is performed on the 8th solution snapshot.

In the NS-MIX experiment, we present randomly selected samples from all test distributions, including the inputs, ground truth solutions, predictions, and corresponding absolute errors. While the predictions for NS-Sines and NS-Sines Moderate may not appear highly inaccurate at first glance, the diffusion-based certificate successfully identified them as OOD, since their errors are significantly larger compared to other distributions. This is also evident in the absolute error plots, where large values occur only for NS-Sines and NS-Sines Moderate. For details, see Figure 23.

In the NS-PwC experiment, we additionally show randomly selected samples from all the test distributions (see 24). While the solutions from the NS-PwC and NS-Brownian distributions are well approximated, the trained model fails to generalize to the other four distributions, whose samples are classified as OOD.

### B.2.1 INSUFFICIENCY OF $p(x)$ AS CERTIFICATE

In the Table 5, we show the accuracy rates and other metrics for baselines (defined in C) on the NS-MIX problem, based on estimating the input distribution $p(x)$ alone. We observe that all certificates derived from such a task-agnostic approach completely fail on this dataset. This result highlights the necessity of a joint-distribution-based approach for obtaining reliable certificates.

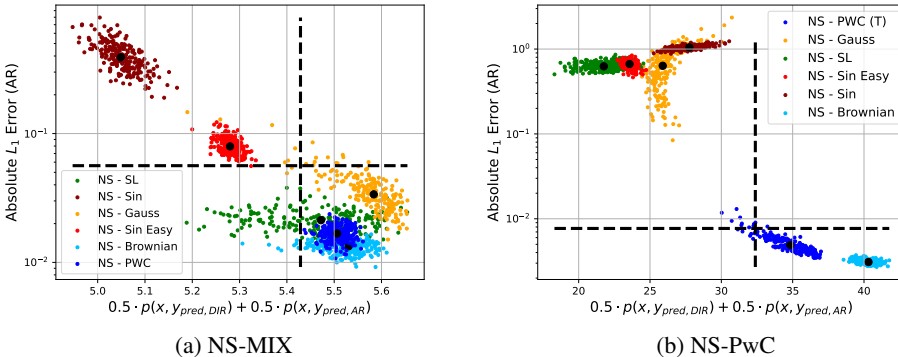

(a) NS-MIX                  (b) NS-PwC

Figure 19: Absolute L1 error versus likelihood for four experimental settings. Each plot shows results for different testing distributions, with the vertical dashed line indicating the likelihood threshold and the horizontal dashed line indicating the error threshold. These decision boundaries divide the space into four quadrants corresponding to true positives, false positives, true negatives, and false negatives.

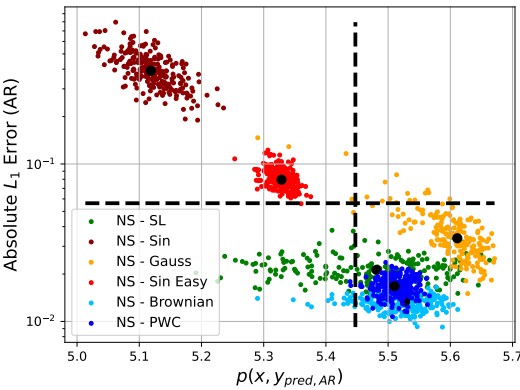

Figure 20: Ablation Study for the certificate. Training distribution is NS-MIX.

### B.2.2   ABLATION STUDY ABOUT THE EVALUATION OF THE LIKELIHOOD

If the models are evaluated autoregressively (AR), our certificate for the time-dependent problems is evaluated as

$$s(x) = 0.5 \cdot p(x, y_{DIR}) + 0.5 \cdot p(x, y_{AR}),$$

where $y_{AR}$ is the AR prediction, while $y_{DIR}$ is the prediction obtained by directly approximating the solution at the test time $T$. We do not use only the $p(x, y_{AR})$, as the model is not trained to make predictions in autoregressive manner. The model is trained to directly predict the solution, so $y_{DIR}$ is the real indicator of how well and accurate our model performs. In the paper Herde et al. (2024), the authors noted that AR evaluation is sometimes beneficial for the model performance, but it is unclear *when* this strategy leads to better performance. For NS-MIX and NS-PwC, we use uniform AR rollouts, using 7 AR steps, with the final evaluation corresponding to the 8th solution snapshot.

Let us now test $s_{AR}(x) = p(x, y_{AR})$ as our certificate. For sufficiently complex training distributions, such as *NS-MIX*, $s_{AR}(x)$ is good certificate, as seen in Figure 20.

However, $s_{AR}(x)$ is not always the best possible indicator. Take for example NS-Sines Moderate training distribution. We trained a regression and a diffusion models on $4.5\mathrm{K}$ trajectories of length 8. If $s_{AR}(x)$ is used, some of the samples that have larger than $20\%$ relative error are classified as in-distribution. The mixed certificate is **more conservative**, as it punishes the model's inability to directly predict the solution (with one forward pass). In Figure 21, we show the performance

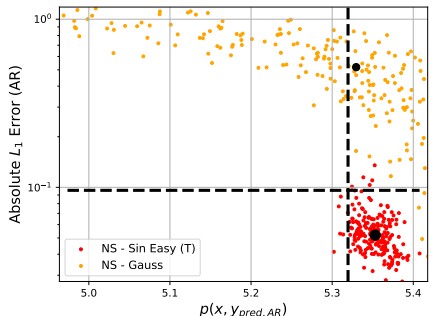 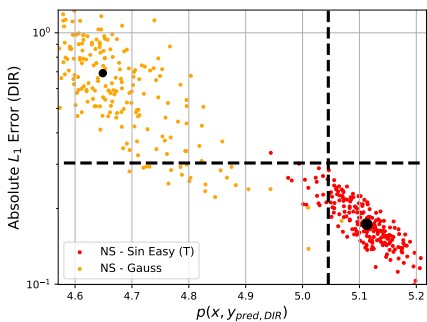

Figure 21: Ablation Study for the certificate. Training distribution is NS-Sin-Moderate. Left: AR Evaluation with the certificate $p(x, y_{AR})$. Right: Direct Evaluation with the certificate $p(x, y_{DIR})$.

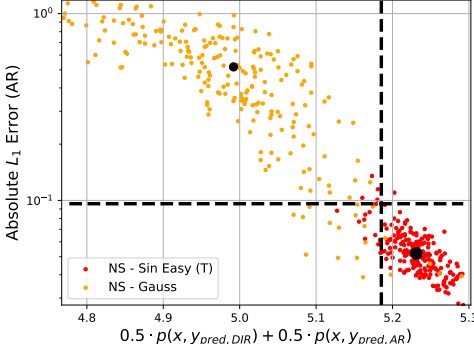

Figure 22: Ablation Study for the certificate. Training distribution is NS-Sines Moderate. Evaluation with mixed certificate.

of the certificate $s_{AR}(x)$. In the same figure, right, we show the error of the direct evaluation vs $p(x, y_{DIR})$. We see that the model is generally unable to accurately predict the solution with direct evaluation in case of NS-Gauss. Thus, we cannot expect the performance of the AR evaluation to be accurate, either. The results of the *mixed* certificate are shown in Figure 22.

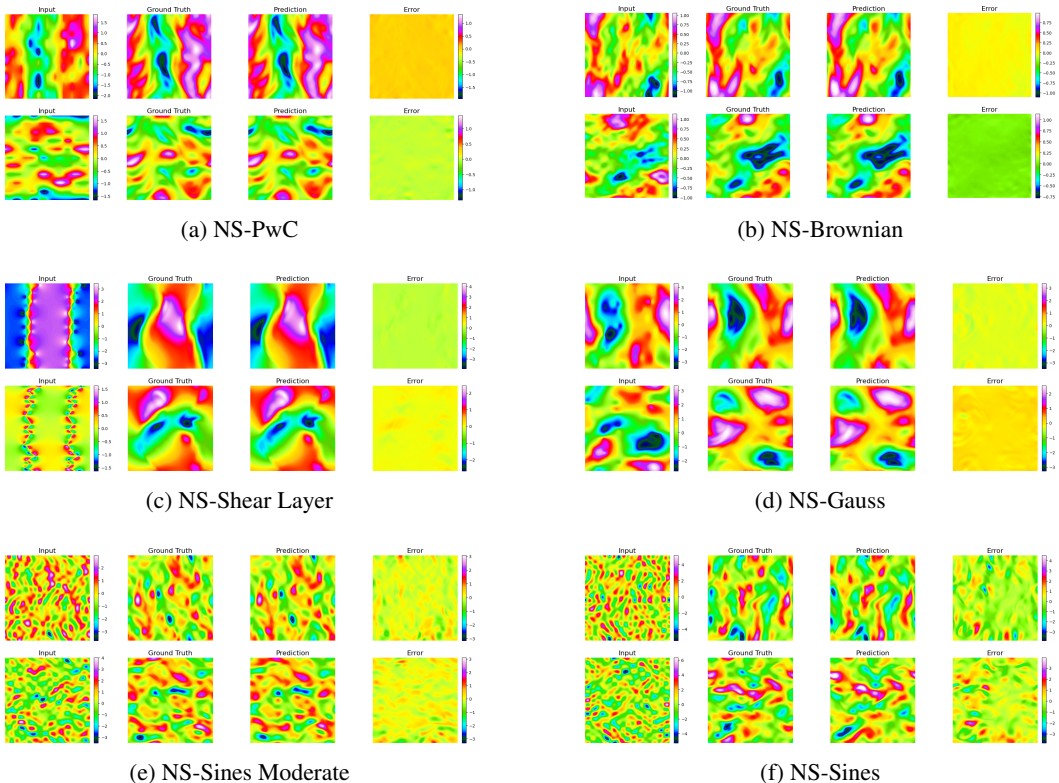

Figure 23: Randomly selected samples from the testing distributions in NS-MIX experiment, showing inputs, ground truth solutions, model predictions, and corresponding absolute errors. While predictions for NS-Sines and NS-Sines Moderate appear visually reasonable, their significantly larger errors compared to other distributions allow the diffusion-based certificate to correctly flag them as OOD. Note that the ground truth outputs, predictions, and absolute errors have the same colorbar.

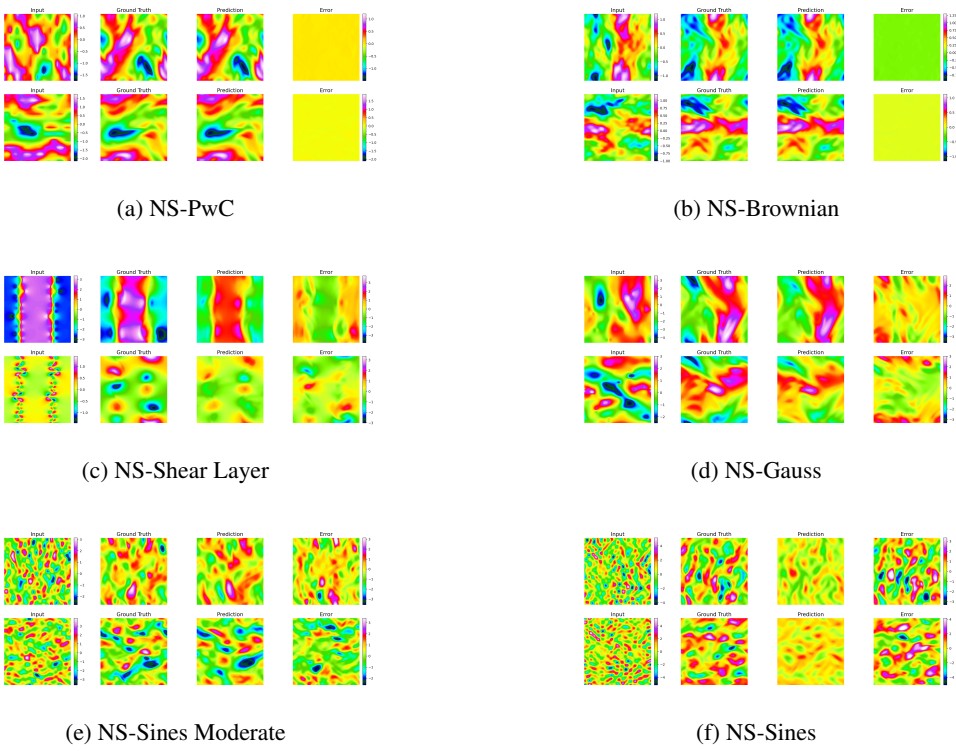

Figure 24: Randomly selected samples from the testing distributions in NS-PwC experiment, showing inputs, ground truth solutions, model predictions, and corresponding absolute errors. The predictions for NS-PwC and NS-Brownian appear visually accurate, while the remaining distributions exhibit larger errors. Our method successfully identifies these other distributions as OOD. Note that the ground truth outputs, predictions, and absolute errors have the same colorbar.

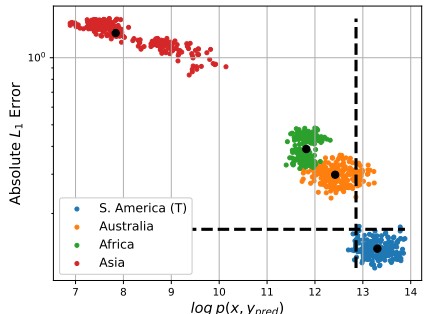 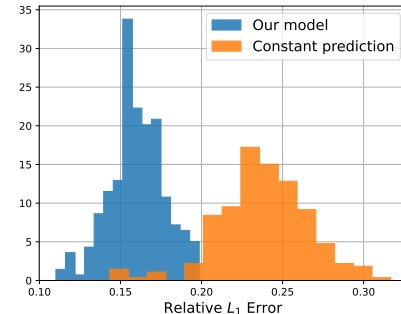

Figure 25: Humidity prediction. Left: $L_1$ errors vs. Estimated log likelihood for different testing datasets. Right: Histogram of absolute errors for 12-hour humidity predictions. The comparison is between our trained model and a persistence forecasting baseline, which assumes no change in humidity over time.

## B.3 HUMIDITY FORECAST

In this experiment, we use MERRA-2 satellite data to forecast surface-level specific humidity over various global regions, in a period January–April. Refer to Figure 26 for an illustration of the data format. The objective is to predict specific humidity **12 hours into the future**. We evaluate our models on humidity prediction for the year 2023 using four different test datasets:

1. South America - Training region
2. Australia and Oceania region
3. African region
4. Asian region

Since humidity patterns vary significantly across continents, we expect poor performance in regions that differ from the training domain. Figure 25 presents $L_1$ errors plotted against the estimated log-likelihood $p(x, y_{pred})$, where $y_{pred}$ denotes the 12-hour predicted humidity. We observe that the diffusion model assigns high likelihoods (corresponding to low prediction errors) to samples from South America. Samples from Australia get slightly lower likelihoods and are mostly identified as OOD. As expected, the African and Asian datasets fall entirely within the OOD region.

We observe that the predicted humidity appear too smooth, lacking fine-scale structures. This is expected, as capturing small-scale features is challenging without providing additional information (such as boundary conditions, or auxiliary variables like wind speed, air temperature, pressure, etc). In fact, our regression task is mathematically ill-posed, so perfect predictions are not expected. In Figure 25 (Right), we compare the error histogram of our model's 12-hour humidity predictions with *persistence forecasting* baseline, where the humidity is assumed to be constant over time (the output is identical to the input). The comparison shows that our model clearly outperforms the persistence baseline. This is evidenced by the error distribution of our model being shifted to the left. Note that all the statistics are computed over **normalized** data.

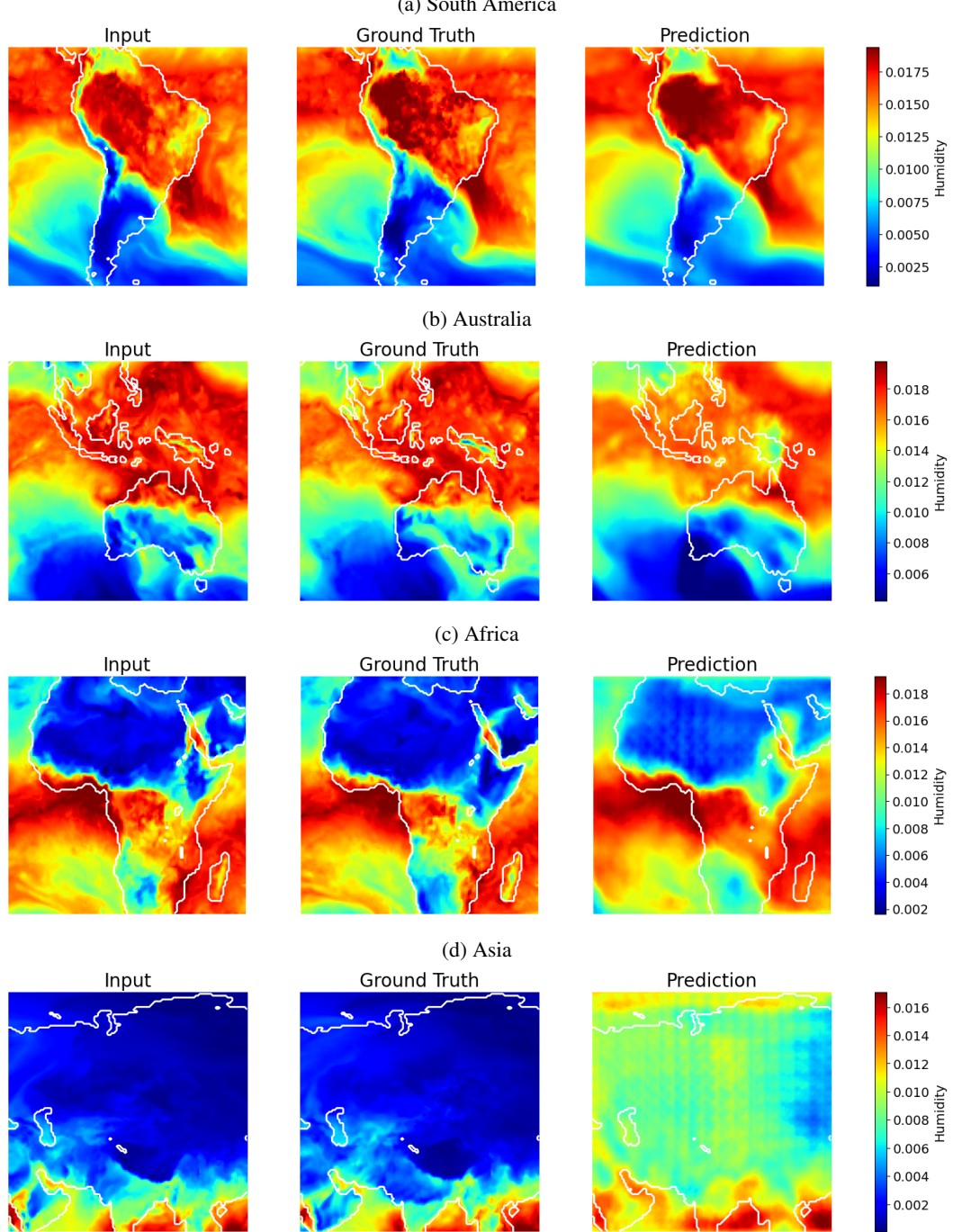

Figure 26: Humidity prediction over different testing regions.

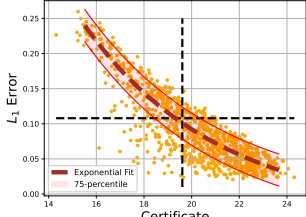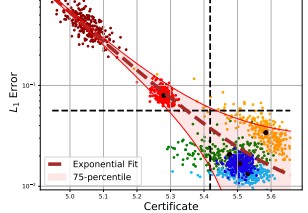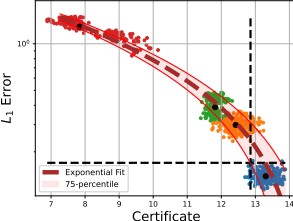

Figure 27: Fitted exponential curves of regression error as a function of the certificate (estimated log-likelihood) for the Wave Equation, NS-MIX, and MERRA2 test cases. Shaded regions denote the 75th-percentile deviation regions, within which the majority of test samples are contained.

### B.4 A POSTERIORI ERROR ESTIMATES

Once the regression and diffusion models are trained on the training distribution, and a set of **test** samples is available, the prediction error can be observed as a function of the certificate values (in our case, the estimated log-likelihood). We illustrate this relation in three settings: Wave Equation, NS-MIX, and MERRA2. We assume the availability of approximately 64 samples (i.e. input-output pairs) from the test distribution for constructing the error–certificate curve. For the Wave Equation, all 64 samples come from a single test distribution. In the NS-MIX case, with six test distributions, we take 11 samples from each (66 in total). For MERRA2, which has four test distributions, we consider 16 samples per distribution (64 in total).

We compute the $L_1$ errors of the regression model on the available test samples and estimate the corresponding certificates. A parametric exponential function of the form

$$y(x) = a \cdot \exp(-bx) + c$$

is then fitted to the certificate–error pairs. Figure 27 presents the fitted curves for the Wave Equation, NS-MIX, and MERRA2 experiments. From each set of samples, we evaluate the absolute deviation between the fitted curves and the true errors, and calculate the 75th percentile of these deviations. Majority of the test samples are contained within the 75th-percentile bands.

### B.4.1 INFERENCE ON TRAINING DISTRIBUTION

In certain situations, the goal is to evaluate how well the model generalizes within its own training distribution. The challenge in this setting is to identify the "most difficult solutions" that lie inside the training distribution. In such cases, one can also perform a posteriori error estimates. We carry out these estimates on the training distributions of the Wave Equation and NS-PwC experiments. A set of 64 samples from the training distribution is used to determine both the likelihood and the error bounds, as well as to fit the exponential relationship between the certificate and the error. To establish the uncertainty bounds, we apply the 75th-percentile rule. We present the error fits alongside the corresponding error–certificate histograms for the Wave-Eq and NS-PwC experiments in Figure 28

**Ablation of threshold.** In the previous cases, the uncertainty bands were defined using the 75th percentile of the absolute error deviations as the threshold. We now vary this threshold and illustrate how the uncertainty bounds evolve as the threshold value increases. Figure 29 shows this evolution for the 65th, 75th, 85th, and 95th percentile bounds for the Wave-Eq experiment. We find that at the 75th percentile, the vast majority of samples lie within the bounds while the underlying uncertainty remains moderate. At the 95th percentile, nearly all samples are contained within the bounds (at the cost of significantly larger uncertainty).

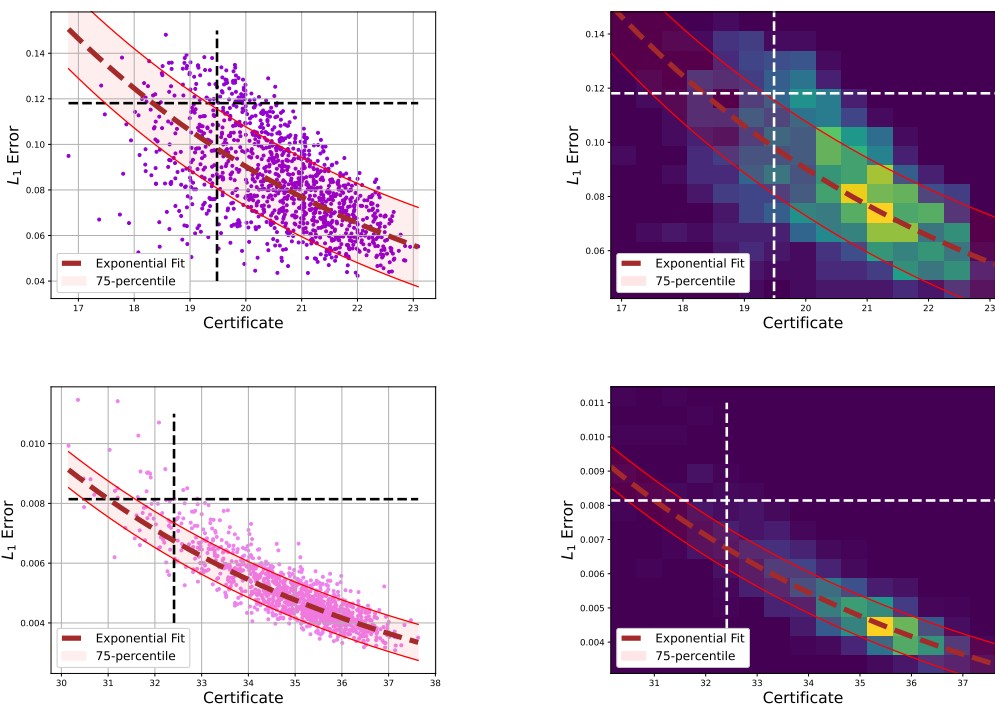

Figure 28: Error fits and corresponding error–certificate histograms for the training distributions. The top panels show results for the Wave Equation, while the bottom panels correspond to the NS-PwC experiment.

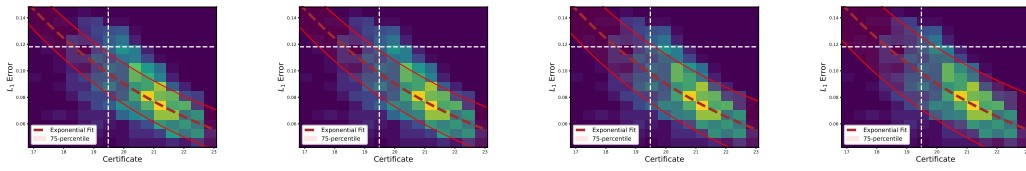

(a) 65-percentile      (b) 75-percentile      (c) 85-percentile      (d) 95-percentile

Figure 29: Evolution of the uncertainty bounds for thresholds set at the 65th, 75th, 85th, and 95th percentiles. Lower thresholds (e.g., 75th) capture most samples with moderate uncertainty, while higher thresholds (e.g., 95th) capture nearly all samples but result in larger uncertainty.

### B.5 IMAGE CLASSIFICATION

Let $x$ be an image with $c$ channels (where $c = 3$ for RGB images and $c = 1$ for grayscale images). Let $y$ be the label associated with that image. The goal of a classification model $\Psi$ is to predict the label $y$ of the image $x$. In probabilistic terms, it is challenging to work with $p(x, y_{true})$ and interpret $p(x, y_{pred})$ in a continuous sense, since $y$ is a discrete label. Although the predicted label is discrete, the model $\Psi_\varphi$ is trained using a *softmax*-based loss function, which assigns log-probabilities to all possible labels and maximizes the log-probability corresponding to the true label $y$.

On top of the classifier, we train a diffusion model to predict the joint probability function $p(x, y)$. **During training**, we concatenate an additional channel containing the constant value $y$ to the $c$ channels of the image $x$. **During testing**, if we only use predicted label $y_{pred}$, we are not fully leveraging the output of the classifier. In that case, only the label corresponding to the highest log-probability would be used in the likelihood estimation. Relying only on the predicted class does not capture the confidence of the classifier in its prediction. To address this, we define the predicted label as a function of the full set of log-probabilities produced by the classifier, represented by the last layer (before the softmax is applied).

Let $M$ be the number of classes and $(l_2, l_2, \ldots, l_M)$ be the corresponding log-probabilities. Let $m \in \{1, \ldots, M\}$ and let us define the probability $p_m$ as a softmax applied to the log-probabilities, that is,

$$p_m = \frac{\exp(l_m/T)}{\sum_{k=1}^{M} \exp(l_k/T)},$$

where $T$ is the temperature parameter that we set to $T = 1$. Let $s$ be the resolution of the image, i.e. each channel of the image is in $\mathbb{R}^{s^2}$. Instead of assigning a constant value for each pixel of the label channel, we observe pixels as single realizations of i.i.d. random variable that follow discrete distribution over $\{1, 2, ..., M\}$, with associated probabilities $[p_1, p_2, \ldots, p_M]$. In this way, predictions with low confidence introduce variability into the label channel, effectively "corrupting" those samples. Consequently, samples for which the classifier is confident remain mostly unaffected. By incorporating uncertain label values, we effectively *perturb the one-dimensional manifold* on which the labels reside.

Note that the classifier can predict wrong label with high confidence, but our hope is in the following:

- This does not happen often. The classifier is *usually uncertain* about OOD samples.

- The diffusion model itself understands that some label is wrongly predicted (i.e. classifier predicted a *bird* instead of a *truck*).

### B.6 CIFAR10

In this experiment, we train both a classifier and a diffusion model using the CIFAR dataset, which contains 10 distinct classes. As described in the main text, we designate the class "trucks" as the OOD class. In Figure 31, we show the predicted labels passed to the diffusion model together with estimated log-likelihoods. Some of the labels are uncorrupted, while some are very noisy. We observe the following:

- The classifier is *rarely overconfident in the wrong class*.

- Even when the classifier is overconfident in the wrong class (e.g. the truck in the first row), the estimated likelihood is still much lower than the ones obtained when the classifier is confident in the correct label.

### B.7 MNIST

For the MNIST, we do the same experiment. The OOD class is the *number 9*. Note that the classification task is very easy, so almost all the ID samples are properly classified. Figure 33 shows the predicted labels passed to the diffusion model together with estimated log-likelihoods for this task.

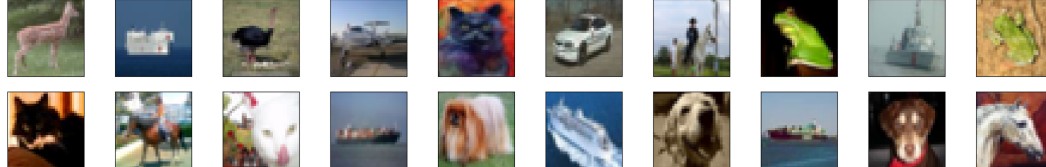

Figure 30: CIFAR Dataset. Samples from the dataset.

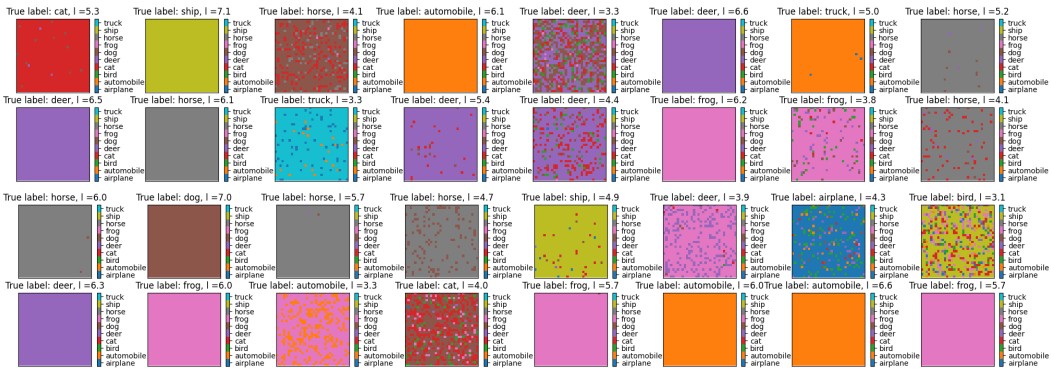

Figure 31: CIFAR Dataset. Labels passed to the diffusion model

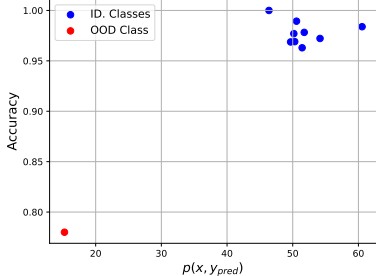

Figure 32: MNIST Image Classification. Accuracy vs. Likelihood Certificate.

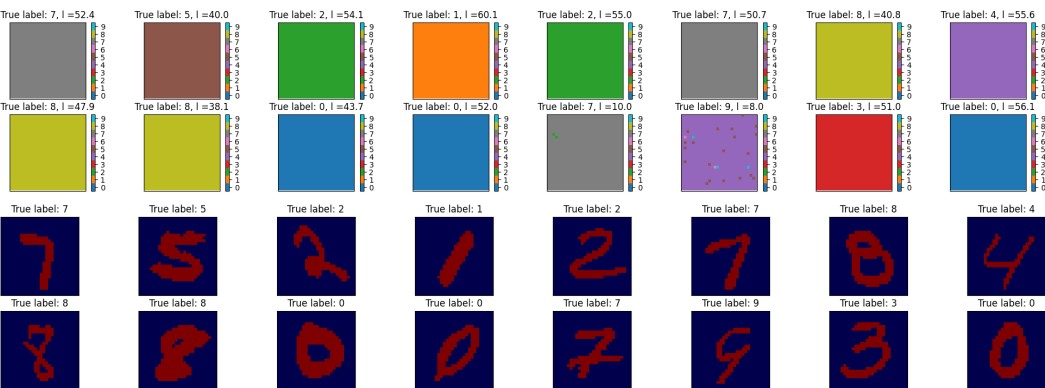

Figure 33: MNIST Dataset. Up: Labels passed to the diffusion model. Down: Samples to be classified.

### B.7.1 Ablation Studies

In Ren et al. (2019), the authors investigated OOD detection using likelihood-based methods. They found that non-semantic pieces of information (such as background pixels in natural images) can substantially affect likelihood estimates, sometimes leading to incorrect OOD decisions. For example, a likelihood-based model trained on the CIFAR-10 dataset may assign higher likelihoods to samples from the SVHN dataset, despite never seeing them during the training.

In our setup, the first three channels represent the image, while the fourth channel encodes the image label/class. Although the class is originally a discrete scalar, we embed it into a much higher-dimensional space ($32^2$-dimensional space in our benchmarks). Similarly to background pixels in natural images, this low-entropy, high-dimensional embedding can significantly influence likelihood estimation. As explained in the previous section, we avoid this effect by perturbing the label/class channel. By introducing noise and incorporating information from the output of the trained classifier, we increase the entropy of the label channel, which may reduce its dominance in the likelihood estimation process.

To show the effect of the perturbations, we now use **unperturbed** labels for likelihood estimation. We assign the pixel values of the class channel to the class with the highest predicted probability from the classifier. In the left panel of Figure 34, we show the classification accuracy of each label against the median estimated likelihood of the corresponding samples. We observe no clear correlation between accuracy and estimated likelihood. Notably, the most accurate class exhibits the lowest likelihood, which differs from the behaviour observed when the noisy labels were used. This result clearly suggests that the estimated likelihood is heavily influenced by background pixels and the unperturbed label channel. To further support this hypothesis, we evaluate our models on the SVHN dataset, which contains images of street view house numbers (see 36 for SVHN samples). Since the classifier has never been exposed to SVHN labels during training, its predictions are **always** incorrect. However, we can still estimate the joint likelihood $p(x, y_{pred})$ using the diffusion model, where $y_{pred}$ denotes the unperturbed, predicted labels. In the right panel of Figure 34, we compare the histograms of estimated likelihoods for the CIFAR-10 and SVHN datasets. We observe that the SVHN samples exhibit generally higher likelihoods than those from CIFAR-10. This failure mode has also been documented in Nalisnick et al. (2019b).

We now evaluate our setup using noisy labels (*NL* abbr.) for likelihood estimation. In Figure 35a, we compare the histograms of estimated likelihoods for all samples in the CIFAR-10 and SVHN test sets. Unlike the previous results with unperturbed labels, the SVHN samples no longer show higher likelihood estimations. However, the two distributions now clearly overlap.

To improve OOD detection, we sequentially refine the subset of CIFAR-10 samples used for generating likelihood histograms. In Figure 35b, we restrict the analysis to **correctly classified** CIFAR-10 samples (*CC* abbr.). Figure 35c shows the histograms for **high-confidence** samples (*HC* abbr.). Those HC samples are the ones for which the classifier assigns at least $90\%$ confidence to some class . Note that in general, the HC class is not necessarily the accurate one. Finally, in Figure 35d, we focus on samples that are both HC and CC.

In these last two figures (NL+CC and NL+HC), the CIFAR-10 likelihood histogram shifts to the right, creating a clearer separation from the SVHN distribution. Next, we select a subset of CC and HC CIFAR-10 samples (NL+CC+HC). By defining a threshold around the median of the estimated likelihoods (within one standard deviation, for instance), we can successfully detect a large portion of SVHN samples as OOD. This whole analysis further supports our assumption that the classifier is rarely overconfident in incorrect labels (provided that the true label belongs to the classifier's label space). Further separation between SVHN and CIFAR-10 likelihoods requires decreasing the influence of non-semantic, background pixels on the likelihood estimates (as proposed and demonstrated in Ren et al. (2019)).

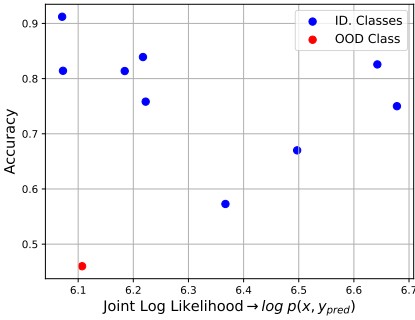 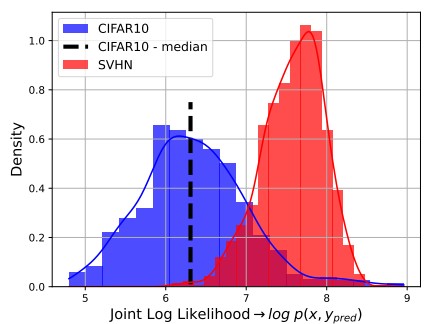

Figure 34: Left: Median estimated likelihood vs. classification accuracy for each CIFAR-10 class using unperturbed labels. Right: Histogram of estimated likelihoods for CIFAR-10 and SVHN samples; SVHN exhibits higher likelihoods despite being OOD.

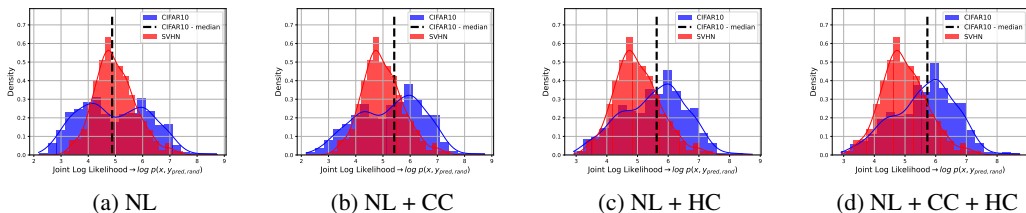

| (a) NL | (b) NL + CC | (c) NL + HC | (d) NL + CC + HC |

Figure 35: Histograms of estimated likelihoods for CIFAR-10 and SVHN test samples under different filtering strategies of CIFAR-10 dataset. (a) All test samples using noisy labels (NL). (b) Only correctly classified (CC) CIFAR-10 samples. (c) Only high-confidence (HC) samples (d) Samples that are both correctly classified and high-confidence (CC + HC). As the selection becomes more refined, the CIFAR-10 likelihood distribution shifts to the right, improving separation from SVHN.

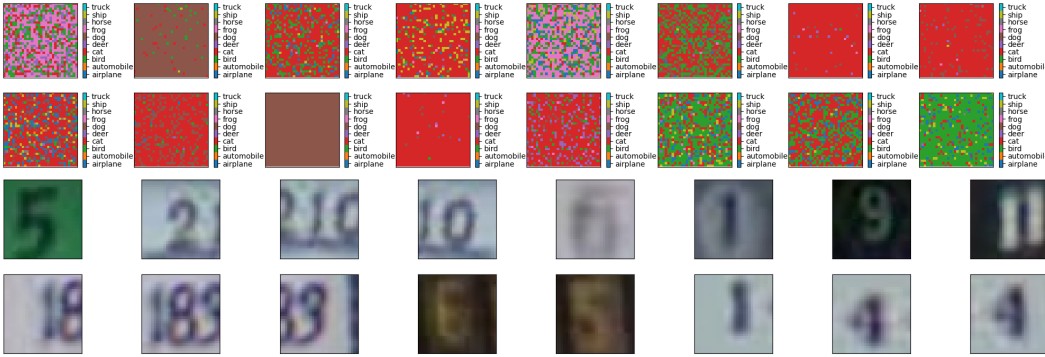

Figure 36: SVHN Dataset. Up: Labels passed to the diffusion model. Down: Samples to be classified.

B.8    BRAIN TUMOR SEGMENTATION

This section contains additional results for the evaluation of our approach on **binary segmentation** tasks. Since the segmentation is nothing but pixel-wise classification, our method follows a similar strategy to the one used for classification tasks. Since in the case of segmentation, we know exactly what the background (non-semantic pixels) are, we explicitly reduce their influence by corrupting them with white noise during the training. We will explain the method after we explain our datasets.

Our objective is to perform **brain tumor segmentation** on the **BraTS2020** dataset. This dataset contains 3D brain MRI volumes of the shape $240 \times 240 \times 155$. The data is divided into two tumor categories:

- High-grade gliomas (HGG)
- Low-grade gliomas (LGG)

Each brain scan also has a multi-class segmentation mask with the following label assignments:

- 0: background
- 1: necrotic core
- 2: edema
- 3: enhancing tumor

For our task, we convert these multi-class masks into binary masks. The transformed labels are defined as:

- 0: non-tumor (background)
- 1: tumor (any of the original classes 1, 2, or 3)

We train our segmentation model using brain scans with HGG tumors. The dataset consists of 210 HGG brain volumes, from which we select 190 for training, 10 for validation, and 10 for testing. To extract 2D slices, we sample 100 slices per brain along the z-axis (corresponding to slice indices 30 through 130). Each slice is resized to a resolution of $128^2$. The pixel values of each brain slice are normalized to $[0, 1]$. During training, we apply a range of augmentation techniques, including horizontal and vertical flips, random rotations, and random shifts and scalings. The input images are **FLAIR** MRI scans.

In parallel with the segmentation model, we also train a diffusion model on the same dataset. During the diffusion training, we exclude the rotation-based augmentations. The diffusion model is trained on the joint distribution $p(x, y)$, where $x$ represents the $2d$ MRI scan of the brain, while $y$ is the binary segmentation mask. Our evaluation is done on 10 held-out HGG brains and an additional set of 10 LGG brains. For the HGG cases, we evaluate the model not only on FLAIR MRI scans, which were used during training, but also on $T_2$-weighted scans. The $T_2$ scans represent a different MRI modality. For the LGG cases, performance is assessed on both axial (z-axis) slices used during the training, and x-axis slices, representing a side view of the brain. This allows us to test the model's generalization to previously unseen anatomical orientations/slices. Note that we test our approach on the brain slices with at least $0.3\%$ tumor pixels present (i.e. at least 50 pixels). For the segmentation model backbone, we use a CNO architecture Raonic et al. (2023) with *silu* activation function.

To reduce the impact of non-semantic (i.e. background) regions, we apply masking during diffusion model training. We replace the background pixels with low-variance Gaussian noise sampled from $\mathcal{N}(0, 0.025)$. During inference, no perturbation is applied. We also present an ablation study in which the diffusion model is trained on unperturbed data for comparison.

Figure 37 shows the relationship between the relative $L_1$ error on the segmentation masks and the estimated log-likelihood of $p(x, y_{\text{pred}})$ for the four test scenarios described earlier. We define the OOD threshold as the *median of the estimated log-likelihoods* computed over the HGG in-distribution test set. Most cases with low segmentation errors are correctly classified as ID. Notably, the vast majority of cases where the segmentation model either predicts an entirely empty tumor mask or produces a mask that has no overlap with the ground truth (i.e., relative $L_1$ error $\geq 1.0$) are correctly identified

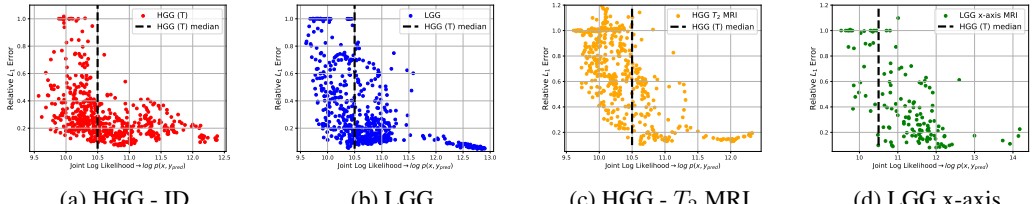

| (a) HGG - ID | (b) LGG | (c) HGG - $T_2$ MRI | (d) LGG x-axis |

Figure 37: Scatter plots showing the relationship between relative $L_1$ error and estimated log-likelihood for test samples across different brain MRI datasets. (a) HGG dataset representing ID samples (b) LGG dataset (c) HGG samples from a different MRI modality ($T_2$ MRI) (d) LGG samples plotted along the x-axis. The plots illustrate how low likelihood values generally correspond to higher errors (OOD), while higher likelihoods align with lower errors (ID).

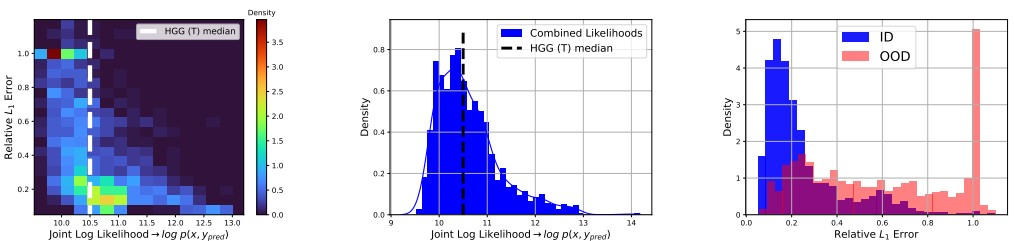

Figure 38: Histograms illustrating the relationship between segmentation quality and model likelihood across various test cases. (Left) Joint distribution of the relative $L_1$ segmentation error and the estimated log-likelihood $\log p(x, y_{\text{pred}})$. (Middle) Distribution of log-likelihoods across test samples. (Right) Distribution of segmentation errors across the same test samples as in the middle. These plots demonstrate that low-likelihood samples often correspond to poor segmentation quality and can be effectively identified as OOD.

as OOD. It is crucial to highlight that our approach effectively identifies OOD samples that come from a **different MRI modality**, namely $T_2$ MRI scans (refer to the third plot in Figure 37).

We now combine the test samples from all datasets. In Figure 38, the first plot shows a 2d histogram of relative $L_1$ error vs. estimated log-likelihood. We first note that the region of highest density correspond to low likelihood values and errors close to $1.0$. Those are correctly classified as OOD. Additionally, many low-error points lie near the classification threshold, but are classified as ID. The second plot shows the histogram of estimated log-likelihoods, which is skewed to the right. In the final plot, we show histograms of errors for samples classified as ID (in blue) and OOD (in red). The ID histogram contains most of the low-error samples, while the OOD histogram includes some low-error samples, but mostly consists of high-error ones. Note that the OOD error histogram has a peak around error $1.0$, indicating that the **most completely incorrect** predictions are classified as OOD.

### B.8.1 ABLATION STUDY - NOISE INJECTION

We now retrain the diffusion model **without adding white noise** to the non-semantic, background pixels during the training. In the left panel of Figure 42, we plot the relative $L_1$ error of the predicted segmentation masks vs. the estimated log-likelihood $\log p(x, y_{\text{pred}})$. The results show that many of the high-likelihood samples correspond to predictions with large errors (i.e., relative error $\geq 1.0$). This indicates that the highest likelihood predictions often correspond to cases where the model fails to detect any tumor, despite it being present in the ground truth. Since the majority of pixels in the segmentation masks represent non-tumor regions, and no noise was applied during training, the model tends to assign higher likelihood to completely non-semantic (no-tumor) predictions.

The right plot of Figure 42 shows the error histograms for ID and OOD samples, (where ID/OOD is defined by the median across all predictions). Unlike the behavior observed when noise was

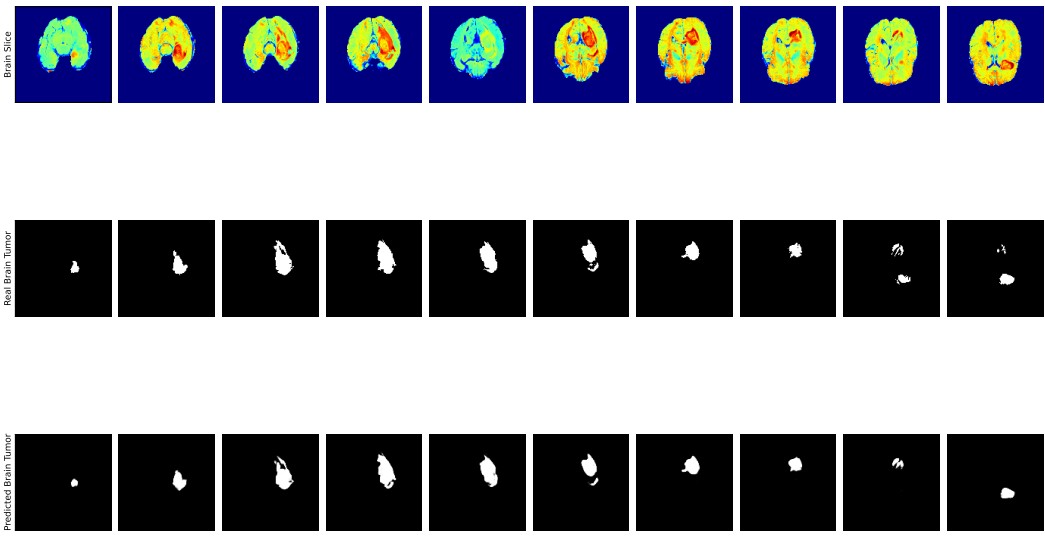

Figure 39: An example of an HGG brain samples (first row), ground truth segmentation masks (second row) and predicted segmentation masks (third row).

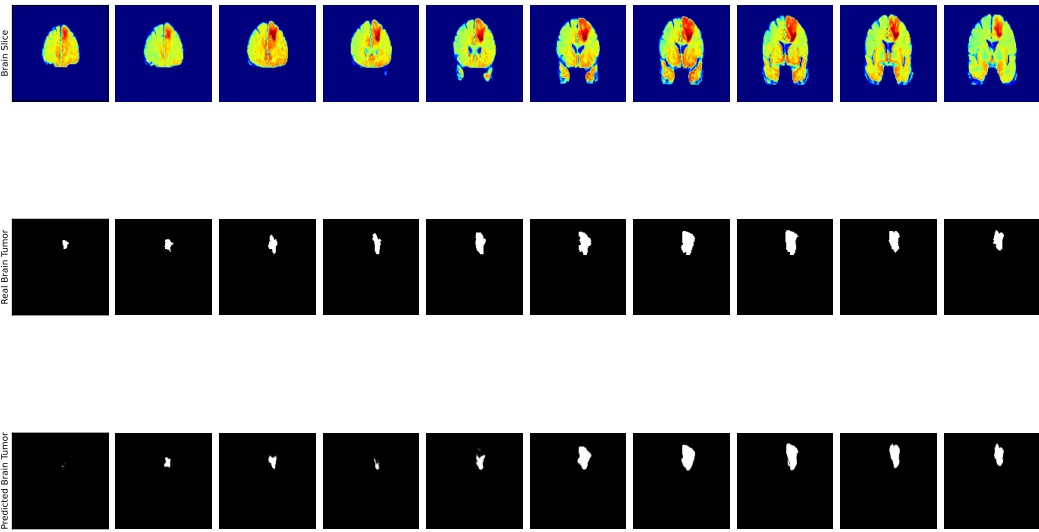

Figure 40: An example of an LGGx brain samples (first row), ground truth segmentation masks (second row) and predicted segmentation masks (thrid row).

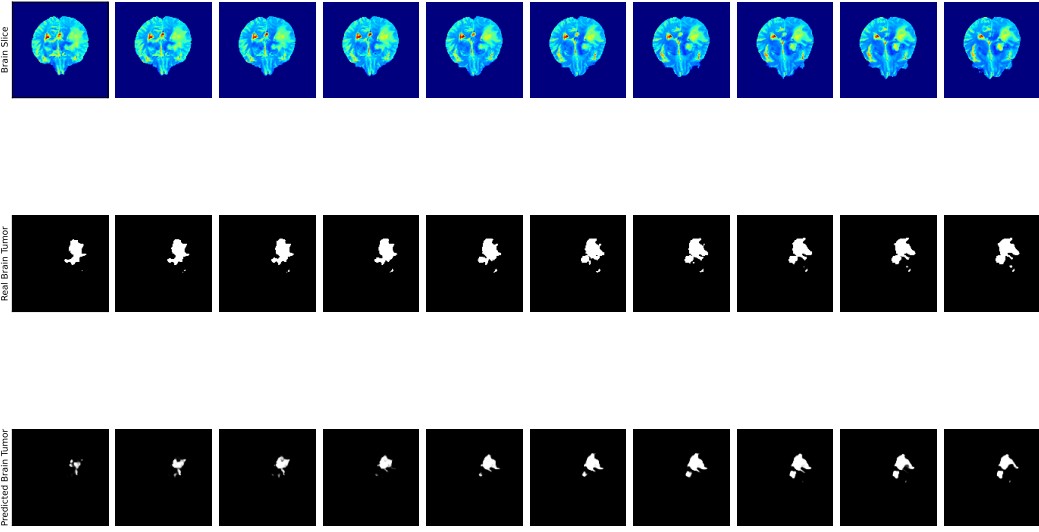

Figure 41: An example of an HGG-T2 brain samples (first row), ground truth segmentation masks (second row) and predicted segmentation masks (thrid row).

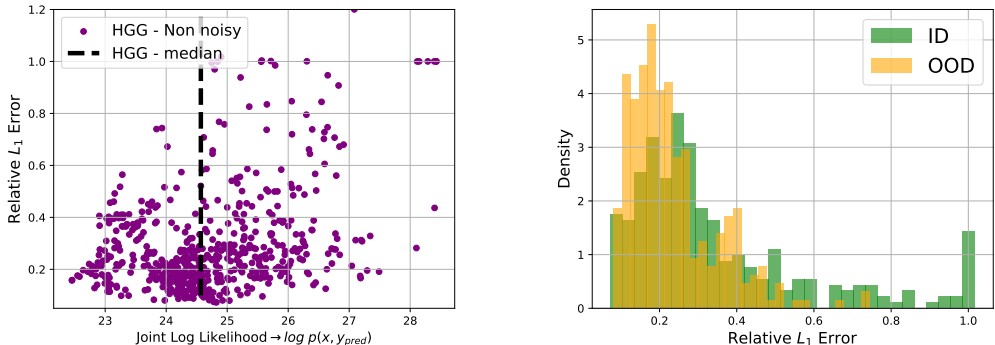

Figure 42: Ablation study: diffusion model trained without noise injection into background pixels. (Left) Relative $L_1$ error of the predicted segmentation masks versus the estimated log-likelihood $\log p(x, y_{\text{pred}})$. High-likelihood samples mostly correspond to large segmentation errors, often representing cases where the model predicts no tumor, despite its presence. (Right) Histogram of segmentation errors for ID and OOD samples. Unlike the noise-injected setting, ID samples now have both low and high error regions, while OOD samples tend to have low errors. This result shows the failure of the method without noise injection.

injected during training, we now see that ID samples have both low and high error values, while OOD samples mostly correspond to low-error cases. This reverse behavior indicates that, without noise injection, the model fails to associate high likelihoods with accurate predictions. This further suggests that noise injection during training is crucial for effective OOD detection.

### B.8.2 CLASSIFICATION SENSITIVITY

In brain segmentation, particular attention must be paid to highly problematic cases, such as:

- No cancer pixels are detected despite their presence.

| Scan type | $\beta_{\mathbf{ERR}}$ | $\alpha_{\mathbf{L}}$ | Accuracy | FPR | FNR | ARCB |
|---|---|---|---|---|---|---|
| HGG | | | 0.675 | 0.003 | 0.322 | |
| LGG | 0.1 | 0.25 | 0.733 | 0.056 | 0.211 | – |
| HGG L2 | | | 0.720 | 0.143 | 0.137 | |
| LGGx | | | 0.844 | 0.106 | 0.005 | |
| average | | | **0.743** | **0.077** | **0.180** | **0.743** |
| HGG | | | 0.530 | 0.002 | 0.468 | |
| LGG | 0.1 | 0.00 | 0.677 | 0.027 | 0.296 | – |
| HGG L2 | | | 0.759 | 0.079 | 0.162 | |
| LGGx | | | 0.670 | 0.041 | 0.289 | |
| average | | | **0.659** | **0.037** | **0.304** | **0.827** |
| HGG | | | 0.787 | 0.028 | 0.185 | |
| LGG | 0.1 | 0.50 | 0.783 | 0.080 | 0.137 | – |
| HGG L2 | | | 0.704 | 0.208 | 0.088 | |
| LGGx | | | 0.816 | 0.142 | 0.042 | |
| average | | | **0.772** | **0.114** | **0.114** | **0.611** |

Table 6: Performance of the segmentation experiments for different values of $\alpha_L$ with $\beta_{\mathrm{ERR}}$ fixed at $0.1$. Results are reported for different scan types (HGG, LGG, HGG L2, and LGGx) in terms of accuracy, false positive rate (FPR), false negative rate (FNR), and accuracy on critical brain cases (ARCB). Average values across different scan types are also provided.

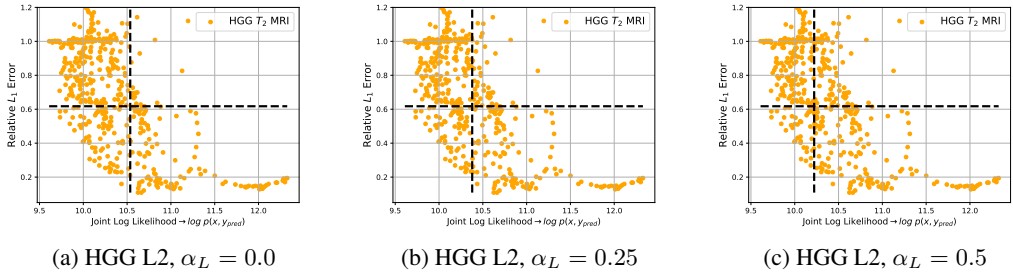

(a) HGG L2, $\alpha_L = 0.0$  (b) HGG L2, $\alpha_L = 0.25$  (c) HGG L2, $\alpha_L = 0.5$

Figure 43: Brain segmentation results for the HGG L2 case. Each plot includes the corresponding likelihood and error decision boundaries, illustrating how the choice of $\alpha_L$ affects the separation of ID and OOD.

- Cancer pixels are detected in a cancer-free brain slice.
- Cancer pixels are present but completely missed, with other pixels incorrectly detected instead.

These situations correspond to relative $L_1$ errors of $\geq 1.0$. A crucial property of the OOD detector is to classify such cases as OOD. For this reason, in the segmentation task, we introduce an additional metric, **ARCB** (*accuracy rate - critical brains*). The ARCB measures accuracy rate specifically for these critical cases.

We empirically observe that the values of $\beta_{ERR}$ (defined in A.4) in the range $(0.1, 0.25)$ yield the most stable performance. For all the following evaluations, we fix $\beta_{ERR} = 0.1$. We then vary the parameter $\alpha_L$ and report the corresponding performance of our method in Table 6. We observe that increasing $\alpha_L$ leads to higher overall accuracy, but also results in a higher FPR and a lower ARCB. This indicates a clear trade-off between maximizing accuracy and maintaining a high ARCB, with a low FPR. The most balanced performance is achieved at $\alpha_L = 0.25$.

Figure 43 shows the brain segmentation results for the HGG L2 case, where each plot presents the corresponding likelihood and error decision boundaries for different $\alpha_L$.

## C    DIFFUSION-BASED CERTIFICATES

Let us observe the probability flow ODE of the form

$$\frac{dx}{dt} = -\frac{1}{2}\sigma_t^2 s(Y_t; t), \tag{27}$$

where $s(Y; t) \approx \nabla_x \log p_t(x)$ is the score function learned during training. As noted in the section A, one estimate the true log-likelihood $\log p(x)$ of a data point $x$, by numerically solving Eqn. 27 backwards in (diffusion) time. We call this approach *Joint Likelihood Based Certificate* (or *JLBC*). We train a score-based diffusion model to estimate $\nabla_x \log p_t(x)$, where $x$ is the **joint distribution** $(x_0, \Psi(x_0))$, where $\Psi$ is the operator of interest.

Once the score function has been estimated, additional probability-flow ODE–based certificates can be constructed. All such certificates originate from the same diffusion model trained on the joint distribution of inputs and outputs. Unlike the classification baseline, diffusion-based methods require no extra samples for OOD detection. Let us define the rescaled score function as

$$\epsilon(Y_t; t) = -\sigma_t \cdot s(Y_t; t).$$

One can define a **unified**, score-based certificate as

$$a(Y) = \alpha \left\| \sum_{t=1}^{S} \epsilon(Y_t; t) \right\|^p + \beta \left\| \sum_{t=1}^{S} \frac{\partial \epsilon(Y_t; t)}{\partial t} \right\|^p + \gamma \sum_{t=1}^{S} \| \epsilon(Y_t; t) \|^p.$$

Here, $t = 1, \ldots, S$ denotes the discrete time steps used in the numerical approximation of the solution of the probability flow ODE (RK steps), and $\alpha, \beta, \gamma \in \{0, 1\}$. The partial derivatives with respect to time (second term in the equation) are approximated with a finite difference scheme. When $\alpha = \gamma = 0$ and $\beta = 1$, the certificate reduces to the curvature-based quantity proposed in Heng et al. (2024) for image classification (referred to as *DiffPath*). Our method differs in that it is trained on the joint distribution, and we therefore refer to it as *JDPath*. When $\alpha = \beta = 1$ and $\gamma = 0$, the certificate incorporates contributions both from the curvature of the score function and from the score function itself. This approach was proposed for medical image classification in Abdi et al. (2025) (termed *SBDDM*). In our joint-distribution, score-based settings, we denote this variant as *JSBDDM*. When $\alpha = 1$ and $\beta = \gamma = 0$, only the contribution from the sum of the score functions remains. We refer to this approach as the *Joint Score Function Norm Score* (*JSFNS*). For $\gamma = 1, \alpha = \beta = 0$. This is a variant of the certificate introduced in Mahmood et al. (2020) is referred to in our framework as *JMSSM*. As noted above, we **unified different certificates** into one single expression, and made them fully operational in our joint-distribution, score-based diffusion settings.

For $M$ testing sample used to define the OOD boundary, we evaluate the different certificates. Then, we compute the median, $m$, together with the standard deviation, $\sigma$. The OOD boundary is defined as

$$l = m + c \cdot \sigma,$$

where $c$ is a tunable parameter, fixed to $c = 1.5$ in all our regression experiments and $c = -0.5$ in the segmentation experiment. Note that the definition of $l$ involves a *plus* sign, in contrast to the likelihood-based approach, since larger errors correspond to larger certificate values (in the regression cases).

# D    MODELS AND ARCHITECTURES

## D.1    CLASSIFICATION BASELINE

We compare our approach against a **classification baseline** that we construct. Specifically, after training a task-specific model, we draw $M$ samples **from the test distribution**. Using the horizontal (error) boundary $e_b$ defined before, we **assign binary labels** to these $M$ test samples. A classification model is then trained on this labeled set, with $0.2 \cdot M$ samples reserved for validation, and $0.8 \cdot M$ for training. Once trained, this classification model is used to predict the ID/OOD classes of the remaining test samples.

In each setting, the classification baseline is trained on **112 test samples** (90 training and 22 validation samples) . For experiments that involve multiple test distributions, we sample an equal number of trajectories from each distribution to construct the training set. If the testing distribution contains $K$ datasets, we use samples from $K/2$ of them to train the baselines. This ensures a fairer comparison with our method. The $M$ samples used for training are excluded from inference. It is important to note that this comparison is unfair, since our method is able to identify ID/OOD samples in a zero-shot manner. This classification baseline **relies on access to $M$ ground-truth solutions** for training, which is the requirement we aim to eliminate. We call this classification-based approach *OODC*.

## D.2    TASK-SPECIFIC MODELS

For all our tasks, we employed the CNO Raonic et al. (2023) architecture with *silu* activations. In all the regression tasks, the model is augmented by *transformer* blocks at selected layers. We refer to this modified design as the *Operator-UViT* architecture. We used architecture of different sizes, depending on the problem. We report the architectural details and the training setups for all the problems in Table 7.

| Setting | Wave Eq. (CNO-Very-Small) | NS-PwC (CNO-Small) | MERRA2 (CNO-Small) | NS-MIX (CNO-Base) | Brain-Segm. (CNO-Small-NoAtt) | MNIST (CNO-Small-NoAtt2) | CIFAR10 (CNO-Small-NoAtt2) |
|---|---|---|---|---|---|---|---|
| **Architecture** | | | | | | | |
| Lifting dimension | 32 | 48 | 48 | 64 | 32 | 32 | 32 |
| # Up/Down layers | 4 | 4 | 4 | 4 | 4 | 4 | 4 |
| Residual blocks (bottleneck) | 4 | 4 | 4 | 4 | 8 | 6 | 6 |
| Residual blocks (middle) | 2 | 2 | 2 | 4 | 8 | 6 | 6 |
| Attention layers used | [T,F,T,F,T] | [T,F,T,F,T] | [T,F,T,F,T] | [T,F,T,F,T] | [F,F,F,F,F] | [F,F,F,F,F] | [F,F,F,F,F] |
| Attention blocks/layer | 4 | 4 | 4 | 6 | – | – | – |
| Attention hidden dim | 256 | 256 | 256 | 384 | – | – | – |
| Attention MLP dim | 256 | 384 | 384 | 512 | – | – | – |
| Attention heads | 4 | 8 | 8 | 8 | – | – | – |
| Attention head dim | 128 | 128 | 128 | 256 | – | – | – |
| Parameters (M) | 21.8 | 41.8 | 41.8 | 113.0 | 17.6 | 11.2 | 11.2 |
| **Training setup** | | | | | | | |
| Optimizer | AdamW | AdamW | AdamW | AdamW | AdamW | Adam | Adam |
| Scheduler | Cosine | Cosine | Cosine | Cosine | StepLP | – | – |
| Initial LR | $10^{-4}$ | $10^{-3}$ | $5 \cdot 10^{-4}$ | $2 \cdot 10^{-4}$ | $5 \cdot 10^{-4}$ | $5 \cdot 10^{-4}$ | $5 \cdot 10^{-4}$ |
| Training samples | 1K | $\sim$140K | $\sim$63K | $\sim$2.97M | $\sim$18K | $\sim$2K | $\sim$40K |
| Epochs | 100 | 100 | 100 | 100 | 100 | 50 | 50 |
| Batch size | 64 | 64 | 64 | 32 | 32 | 96 | 64 |

Table 7: Architectures and training setups across different problems.

## D.3    DIFFUSION MODELS

For the diffusion denoisers $D_\theta$, we adopted the UViT architecture from Molinaro et al. (2025), combined with exponential noise scheduling and a variance-exploding diffusion scheme. Further details are provided in Section 6.3 of Molinaro et al. (2025).

For all the non-classification tasks, we use 4-layer UViT architectures with channel numbers adapted to task difficulty. For classification tasks, where the input resolution is lower, we employ 1-layer UViTs.

For the Wave Equation problems, we use the following architecture of the UViT:

- **Number of layers**: 4
- **Channels per layer**: [32, 64, 128, 256]
- **Number of attention blocks per layer** 4

- **Attention hidden dimension:** 128
- **Attention Heads:** 4
- **Attention Head dimension:** 128
- **Trainable parameters:** 22.2M

For the MERRA2, NS-PwC and the segmentation problem, we use the following architecture of the UViT:

- **Number of layers**: 4
- **Channels per layer**: [48, 96, 192, 384]
- **Number of attention blocks per layer** 6
- **Attention hidden dimension:** 256
- **Attention MLP dimension:** 128
- **Attention Heads:** 8
- **Attention Head dimension:** 128
- **Trainable parameters:** 69.3M

For the classification problems, we use the following architecture of the UViT:

- **Number of layers**: 1
- **Channels per layer**: [256]
- **Number of attention blocks per layer** 4
- **Attention hidden dimension:** 512
- **Attention Heads:** 4
- **Attention Head dimension:** 256
- **Trainable parameters:** 34.0M

## D.4 OTHER REGRESSION MODELS (WAVE EQ.)

The architecture of the UNet model used in the ablation study B.1.4 is:

- **Number of layers**: 4
- **Channels in the layers**: [60, 120, 240, 480]
- **Number of ResNets in the bottleneck**: 2
- **Trainable parameters**: 19.2M

The architecture of the ViT model used in the ablation study B.1.4 is:

- **Number of attention blocks** 6
- **Attention hidden dimension:** 256
- **Attention MLP dimension:** 512
- **Attention Heads:** 6
- **Attention Head dimension:** 64
- **Trainable parameters:** 9.7M

The architecture of the C-FNO model used in the ablation study B.1.4 is:

- **Number of Fourier Layers** 4
- **Number of Fourier Modes:** 16
- **Latent Dimension:** 96
- **Conv. kernels per layer:** $[3, 5]$
- **Trainable parameters:** 19.0M

# E   LLM ASSISTANCE IN WRITING

The LLMs were used only to rephrase certain sentences in the paper. No additional assistance was taken from them in terms of writing.

