# OpenReview forum: "Towards a Certificate of Trust: Task-Aware OOD Detection for Scientific AI"
_ICLR.cc/2026/Conference — ICLR 2026 Poster_

### Official Review · Reviewer_2W7u · 2025-10-25

**Soundness:** 3
**Presentation:** 3
**Contribution:** 2
**Rating:** 6
**Confidence:** 3

**Summary:**

This paper proposes a task‑aware (mostly regression-based tasks) out‑of‑distribution (OOD) detector for scientific ML that estimates a joint likelihood $\log p(x, y_{\text{pred}})$ with a score‑based diffusion model and uses it as a reliability certificate for an arbitrary downstream predictor. The key intuition is a heuristic link between prediction error and joint likelihood (Eq. (2)), and the practical pipeline solves the probability‑flow ODE to score new data points.

Decision thresholds are set from a small set of training samples to separate ID vs OOD (Fig. 1C) and the same score can be used to fit an a‑posteriori error curve when a few labeled test samples are available (Fig. 1D and Fig. 24). Valid experiments are run on canonical datasets: regression on PDE operators (wave and 2D Navier–Stokes), satellite humidity forecasting (MERRA‑2), image classification (MNIST, CIFAR‑10), and brain‑tumor segmentation The proposed method is compared with several diffusion-based baselines on these setups with metrics like ACC, FPR, FDR to demonstrate the effectiveness of this joint likelihood as compared to input-only likelyhood $p(x)$ in correlation to prediction error. Ablation studies are conducted as well to further isolate the impact of number of calibration samples and different task models.

**Strengths:**

1. Empirical result appears to be strong. Fig.2 and Table 3 are good illustration indicative of why $p(x)$ could fail for scientific regression and why joint likelihood is a better proxy.
2. Experiment setups are carefully crafted and elaborated. Full details are presented, and a wide spectrum of data sources are provided with proper justifications.
3. The proposed method is intuitively sound within the context of score-based diffusion, and appears to be robust to various scientific regression tasks.
4. Simple exponential fit provides actionable error estimates once a few test labels are available.
5. Related studies is well discussed, demonstrating the gaps in this field, and strengthening the originality of the work
6. The method shows a certain degree of significance by filling the gaps in good predictors for regressor tasks. However, given the lack of discussion on its computational costs and complexity in real scenarios, it is hard to gauge whether it will be meaningful/provide additional insights for more practical methods in this area.
7. In general, given the depth of the discussion and the breath of the experiments conducted, it is a high quality presentation.

**Weaknesses:**

1. Lack of theoretical supports and further discussions: Equation (2) assumes approximately uniform training error over $p(x,y)$. This key step is not formalized; conditions under which the correlation degrades are not identified (e.g., heavy label noise, multi‑modal conditionals $p(y|x)$ with identical likelihood but disparate errors). Strengthening or bounding the approximation would help.
2. The choice of metrics can be better justified. The metrics used (ACC, FPR, FDR) can be swapped with other common OOD metrics (AUROC, TNR@TPR95) for stronger comparisons to prior works
3. (optional) To strengthen the significance of the work, comparison other baselines other than diffusion-based scores should be considered. For example, (i) likelihood‑ratio methods (e.g., Ren et al., 2019) applied to joint or representation spaces; (ii) ensembles/MC‑dropout as task‑aware uncertainty proxies; (iii) conformal risk‑coverage curves for selective prediction; (iv) energy‑based scores on joint representations.
4. The “median − 1.5σ” rule (Sec. 3; App. A.4) is somewhat ad‑hoc despite ablations (Table 2 & Table 4). More principled calibration (e.g., quantile‑conformal bands or controlling FPR at target levels) would increase reliability.
5. Computing joint log-likelihood requires solving the probability‑flow ODE and estimating divergences with 32 MC samples (App. A.3). Runtime, memory, and scalability versus input/output resolution and time‑steps are not quantified. This matters for high‑res scientific workloads -> Potentially limited practicality and thus significance.
6. Code availability are not provided.

Reference:
Jie Ren, Peter J. Liu, Emily Fertig, Jasper Snoek, Ryan Poplin, Mark DePristo, Joshua Dillon, Balaji Lakshminarayanan. “Likelihood Ratios for Out‑of‑Distribution Detection.” NeurIPS 2019.

**Questions:**

1. On Eq. (2):  Can you formalize conditions under which the error–likelihood correlation provably holds (e.g., bounded density ratio between train/test, Lipschitz loss, coverage of $p(y|x)$? Conversely, can you build a counterexample where $\log p(x, y_{\text{pred}})$ is high but error is high (beyond the anecdotal segmentation failure without noise)?
2. In NS experiments, you average direct and autoregressive likelihoods (Sec. B.2.2, Figs. 18–20). Can you quantify when each dominates and whether the mixture is necessary if the regressor is trained with scheduled rollouts?

---

> ### Author Response · Authors · 2025-11-20
> **Reply to the Reviewer (Part 1)**
>
> We start by thanking the reviewer for their comments and suggestions, for appreciating the strengths of paper and for the constructive criticism in terms of identifying weaknesses/questions that we address in detail below.
>
> [W1.]  **Theoretical guarantees**.  We agree with the reviewer that the theoretical heuristics in Section 2 of the original version were not sufficiently clear. To address this, we now provide a more formal derivation of the rationale behind using joint likelihood estimates as certificates, by establishing an upper bound on the error that explicitly incorporates the likelihood term. Under assumptions, we show in SI A.1 that the (log-)error can be bounded by (see Eqn. 2 in the revised version of the paper (attached))
>     $$
>     \log\left(\ell(y_{{\rm gt}},y_{pred}) \right) \leq \alpha \log(\epsilon) - \log (p(x, y_{pred})) + O(\epsilon^\beta)
>     $$
>     where $\epsilon$ is the average loss and $\alpha,\beta \in (0,1)$. This bound makes explicit the connection between the joint likelihood and the pointwise errors, showing that the likelihood term naturally controls the behavior of the error and thus justifies its use as a certificate. We have provided the detailed derivation and critically examined the underlying assumptions in SI A.1 and hope that it provides a stronger theoretical foundation of our paper.
>
> [W2]: **Choice of the metrics**.  Following the reviewer's excellent suggestion, we report the threshold-independent AUROC metric for all models and baselines in the main results table 1 of the revised version. On average, JLBC attains the highest and most stable AUROC. For the most challenging multimodal setting NS-MIX task, we further include estimated ROC curves in Figure 4, where JLBC demonstrates markedly stronger discrimination compared to the other baselines.
>
> [W3]: **New Baselines** Following the reviewer's excellent suggestion, we considered additional baselines on Bayesian approaches for using epistemic uncertainty for OOD detection,  namely the original MC-Dropout algorithm of Gal et. al and the much more recent and state of the art Rate-in algorithm of Zeevi et. al.  We have compared both these algorithms with our likelihood-based JLBC algorithm for the Wave Eqn. dataset and discussed the detailed algorithms and results in the main text as well as in SI B.1.7 and SI Table 4. As shown in this table, JLBC (ours) is *significantly more accurate* than the best performing Bayesian approach (Rate-in) with an ACC score of $0.87$ (JLBC), compared to $0.76$ (Rate-In). Moreover, JLBC is *much faster to run* at $0.02$ secs, compared to the $0.15$ secs of runtime for the Rate-In algorithm. Thus, we demonstrate that our proposed Diffusion-based approach is significantly superior to the Bayesian approaches. Needless to say, we will add these results into a CRV, if accepted, as they significantly strengthen the conclusions of our paper.
>
> [W4]. **Threshold choices**. We acknowledge the reviewer's point that the median–1.5$\sigma$ rule is somewhat ad hoc, although our extensive ablation studies (Table 2) indicate that it performs consistently and remains stable across settings. Nevertheless, more principled and task-dependent calibration procedures would be desirable. We have incorporated this valuable suggestion into the discussion in the main text.
>
> (Continued Below)

---

> > ### Author Response · Authors · 2025-11-20
> > **Reply to the Reviewer (Part 2)**
> >
> > [W5]. **Computational Overheads**: In response to the reviewer's concern regarding computational cost and scaling of our method. It is essential to note that our method solves the probability-flow ODE in only four internal steps. The Skilling-Hutchinson divergence is estimated using a random tensor of size 32, a value chosen heuristically in the original version. To address its impact, we conducted an ablation study on the tensor size (Wave Eq. experiment). As detailed in SI B.1.6, and SI Table 3,  we found that reducing this size to 2, requiring *only 8* forward passes of the diffusion model, still yields similarly accurate metrics as with size 32. However, with this setup, a  single evaluation of the certificates takes just *0.02 seconds*, and is much faster than the Rate-In runtime. Moreover, we would like to emphasize that the end-user *does not* have to re-train the Diffusion model. It can be used in a *zero-shot* manner, entailing no additional training costs. We would like to thank the reviewer for pressing us to explore this issue, which brings out an added advantage of our method
> >
> > [W6]. **Code Availability**. We would like to point out that the code was already provided in the Supplementary Material of the original submission and be readily accessed from there.
> >
> > [Q1]: We have now clearly formalized the conditions under which Eqn 2 holds. Please see SI A.1 of the revised version.
> >
> > [Q2]: Regarding the reviewer's question on **Rollouts**, we would like too say that in settings with ample training data (e.g., NS-MIX), where the diffusion denoiser generalizes well, both direct and AR likelihoods serve as effective certificates and yield similar results. Differences emerge in data-limited regimes (see ablation in B.2.2): direct predictions produce more stable, conservative scores, whereas AR predictions, being smoother due to multiple forward passes, often receive higher certificates. In these cases, the mixed approach is noticeably more reliable. These effects can be further quantified via posterior estimates, enabling adjusted weighting if desired. Direct evaluations with direct certificates would also be feasible. Finally, if the model were trained with scheduled rollouts, mixed certificates would not be necessary, since training and testing would follow the same procedure (as in the Wave Equation example). However, our model is trained in an all2all fashion (which is a direct prediction approach in its nature), which differs fundamentally from AR training, and this mismatch motivates the need for the mixed certificate approach.
> >
> > We hope that the revised version and our comments address all the concerns of the reviewer. We request the reviewer to kindly upgrade their assessment of our paper accordingly.

---

> ### Comment · Reviewer_2W7u · 2025-11-28
>
> The authors have addressed most of my questions. I'm updating my score to 8

---

> ### Author Response · Authors · 2025-11-28
> **Thanking the Reviewer**
>
> We thank the reviewer for their prompt response and for raising our score to 8. Your comments and feedback have helped us improve the quality of our paper and we are very grateful for it.

---

### Official Review · Reviewer_ARrh · 2025-10-28

**Soundness:** 4
**Presentation:** 3
**Contribution:** 3
**Rating:** 4
**Confidence:** 4

**Summary:**

The authors propose Joint Likelihood-Based Certificate (JLBC), which uses a diffusion model to perform out-of-distribution (OOD) detection. The core idea of this method is to train a diffusion model to approximate the joint distribution $p(x, y_\text{pred})$ where $y_\text{pred}$ is the output of the base regression model. The log-likelihood estimated from the diffusion model serves as a certificate of trust, which correlates with the base model's prediction error. Experimental results over a number of domains (e.g., medical, wave, weather) support the validity of this correlation.

**Strengths:**

1. The method doesn't require any retraining of the base regression model, reducing computational overhead. Moreover, the method operates zero-shot (aka it doesn't require any ground truth OOD vs. ID labels).
2. The use of diffusion models to represent / estimate joint likelihoods is well studied / supported.
3. Empirical studies. The experiments are performed over a number of different datasets and downstream tasks, giving a good idea of the expected performance of the method in real-world scenarios.

**Weaknesses:**

1. Novelty and distinction from prior works is somewhat unclear. Section 2 references a number of OOD detection methods and yet concludes with "[there] is [a] scarcity of OOD detection...methods for...scientific machine learning applications," which seems contrary. Additional discussion specifically on the shortcomings of previous methods---which are addressed by the proposed method---should be included. Moreover, further discussion on diffusion / learning-based methods for uncertainty estimation should be in mentioned, since many of these methods seem relevant (e.g., estimation of epistemic uncertainty is often used for OOD detection).
2. Computational overhead. Diffusion model training and inference can be expensive, perhaps even more so than the underlying regression model.
3. Conciseness. The writing in the manuscript is quite dense (particular the results section) and could benefit from more clear / concise organization.
4. Lack of theoretical guarantees. The method proposes the joint likelihood estimate as a certificate of trust, yet there are no concrete guarantees on the correctness of the estimate (i.e., OOD prediction's) accuracy.

**Questions:**

1. How does the proposed approach compare to uncertainty-based methods (e.g., IGRUE, HyperDM, Rate-in, Bayesian NNs, MC dropout, etc.) on the same tasks?

---

> ### Author Response · Authors · 2025-11-20
> **Reply to the Reviewer (Part 1)**
>
> We start by thanking the reviewer for their comments and suggestions, for appreciating the strengths of paper and for the constructive criticism in terms of identifying weaknesses/questions that we address in detail below.
>
> [W1]: **Novelty wrt Existing Literature**. The framing of the existing literature on OOD detection and error certification of ML models was perhaps unclear in the original version. We agree with the reviewer that numerous NN-based OOD detection methods have been proposed in recent years, However, prior work has focused overwhelmingly on classification in the context of vision/images. On the other hand, OOD detection and error certification in *scientific machine learning* tasks such as physics, weather etc, where the underlying learning objective is regression-based, has been severely under-explored.  The design of OOD detection methods for such scientific applications is the main rationale for our paper. We have more clearly delineated this framing in the revised version (see attached file) Sec. 2.
>
> Moreover as suggested by the reviewer, we have added a short discussion of Bayesian approaches, which use epistemic uncertainty for OOD detection. In addition, we have clearly stated that most of current OOD detection methods use the likelihood of the input distribution. However, this approach leads to very unsatisfactory results (Sec. A.2 and Sec B.2.1). Thus, we had to adapt existing baselines (developed for image classification) to our framework of joint input-output distribution. These contributions are more clearly stated in the revised version and will be added to a CRV, if accepted.
>
> [W2]:  **Computational Overheads**: In response to the reviewer's concern regarding computational cost of our method. It is essential to note that our method solves the probability-flow ODE in only four internal steps. The Skilling-Hutchinson divergence is estimated using a random tensor of size 32, a value chosen heuristically in the original version. To address its impact, we conducted an ablation study on the tensor size (Wave Eq. experiment). As detailed in SI B.1.6, and SI Table 3,  we found that reducing this size to 2, requiring *only 8* forward passes of the diffusion model, still yields similarly accurate metrics as with size 32. However, with this setup, a  single evaluation of the certificates takes just *0.02 seconds*. Moreover and as acknowledged by the reviewer, we would like to emphasize that the end-user *does not* have to re-train the Diffusion model. It can be used in a *zero-shot* manner, entailing no additional training costs. We would like to thank the reviewer for pressing us to explore this issue, which brings out an added advantage of our method
>
> [W3]: **Terseness of the write-up**  We agree with the reviewer that the Section 4 may be difficult to follow. Because a substantial part of our contribution is empirical, spanning many ablation studies on the importance of estimating joint versus input/conditional distributions, regression model variants, decision boundaries, as well as classification and segmentation tasks, it was not possible to include all results within the page limit. The main text presents the key takeaways, while the SI contains the full set of detailed experiments. If the paper is accepted, we will use the additional space to restructure and clarify the presentation of results in the **CRV** (camera-ready version).
>
> (Continued below)

---

> ### Author Response · Authors · 2025-11-20
> **Reply to the Reviewer (Part 2)**
>
> [W4]: **Theoretical guarantees**.  We agree with the reviewer that the theoretical heuristics in Section 2 of the original version were not sufficiently clear. To address this, we now provide a more formal derivation of the rationale behind using joint likelihood estimates as certificates, by establishing an upper bound on the error that explicitly incorporates the likelihood term. Under assumptions, we show in SI A.1 that the (log-)error can be bounded by  (see Eqn. 2 in the revised version of the paper (attached))
>     $$
>     \log\left(\ell(y_{{\rm gt}},y_{pred}) \right) \leq \alpha \log(\epsilon) - \log (p(x, y_{pred})) + O(\epsilon^\beta)
>     $$
>     where $\epsilon$ is the average loss and $\alpha,\beta \in (0,1)$. This bound makes explicit the connection between the joint likelihood and the pointwise errors, showing that the likelihood term naturally controls the behavior of the error and thus justifies its use as a certificate. We have provided the detailed derivation and critically examined the underlying assumptions in SI A.1 and hope that it provides a stronger theoretical foundation of our paper.
>
> [Q1] **New Baselines** The reviewer's suggestion of considering Bayesian approaches which use epistemic uncertainty for OOD detection is excellent. We were not aware of this line of work and thank you for pointing it out. We have now added a discussion of these methods in our Related work Section 2.
>
> More crucially, we have considered two of the suggested baselines for Bayesian approaches, namely the original MC-Dropout algorithm of Gal et. al and the much more recent and state of the art Rate-in algorithm of Zeevi et. al.  We have compared both these algorithms with our likelihood-based JLBC algorithm for the Wave Eqn. dataset and discussed the detailed algorithms and results in the main text as well as in SI B.1.7 and SI Table 4. As shown in this table, JLBC (ours) is *significantly more accurate* than the best performing Bayesian approach (Rate-in) with an ACC score of $0.87$ (JLBC), compared to $0.76$ (Rate-In). Moreover, JLBC is *much faster to run* at $0.02$ secs, compared to the $0.15$ secs of runtime for the Rate-In algorithm. Thus, we demonstrate that our proposed Diffusion-based approach is significantly superior to the Bayesian approaches. Needless to say, we will add these results into a CRV, if accepted, as they significantly strengthen the conclusions of our paper.
>
> We hope that the revised version and our comments address all the concerns of the reviewer, particularly on the comparison to Bayesian approaches. We request the reviewer to kindly upgrade their assessment of our paper accordingly.

---

### Official Review · Reviewer_ToFP · 2025-11-01

**Soundness:** 4
**Presentation:** 2
**Contribution:** 4
**Rating:** 6
**Confidence:** 2

**Summary:**

This work proposes a method to classify test examples as in-distribution or out-of-distribution. To this end, the authors propose a diffusion model to compute the certificate (joint likelihood of test example and the model prediction). The experiments include two PDE datasets, a satellite-data based forecasting, image classification and brain tumor segmentation. Across all datasets, the estimated certificate strongly correlates with prediction errors on ID vs OOD data.

While the results seem to be strong, the results section is really hard to follow with most tables and figures in the Appendix. I'd encourage the authors to move some of the key results to the main text.

**Strengths:**

- The paper studies an important research question and presents strong results across a wide variety of domains and tasks (regression and classification)
- The proposed method works both in a zero-shot setup as well as with a setting with ground-truth examples.
- When compared with other diffusion-based baselines (Table 1), the proposed method outperforms across all datasets.

**Weaknesses:**

- The paper currently misses out on connecting the baselines (lines 425-431) with previously published work. Its unclear if any of these are directly related to the works cited in section 2.
- Its unclear if the gains reported in Table 1 are statistically significant. It would be helpful to include some tests for this.
- The main text (section 4, pages refers extensively to the results in the Appendix, this makes it really hard to follow the arguments. I'd encourage editing the paper to move key results to the main text.

**Questions:**

- Whenever applicable, provide citations for the baselines.
- There is only a short discussion in the paper regarding posterior estimates. I'd encourage expanding this subsection.

---

> ### Author Response · Authors · 2025-11-20
> **Reply to the Reviewer**
>
> We start by thanking the reviewer for their comments and suggestions, for appreciating the strengths of paper and for the constructive criticism in terms of identifying weaknesses/questions that we address in detail below.
>
> [W1.]  **Connecting the baselines**  We agree with the reviewer that the baselines in the original submission were *not* explained clearly. We have now revised the section *Quantitative Performance Metrics and Baselines* (see attached Main text) to provide clearer descriptions and appropriate citations. Since all original baseline methods were *only* designed for a likelihood corresponding to the input-distribution, and perform very poorly for OOD detection (see SI B.1.6, Table 3), we had to adapt them to our joint-distribution likelihood framework to ensure a fair comparison. In addition, we have unified several score-based certificates (see SI Section C) into a single formulation, allowing these methods to be viewed within a common framework.
>
> [W2.] **On Statistical Significance**: We thank the reviewer for the valuable comment regarding the statistical significance of our results. Following their excellent suggestion, we have reported the threshold-independent AUROC metric for all models and baselines in the main table of the revised version (attached here). We observe that JLBC achieves the highest and most stable AUROC on average and for most of the datasets. For the most challenging multimodal setting in the NS-MIX experiment, we additionally provide ROC curves in Figure 4 of the revised main text, where JLBC shows substantially stronger discrimination compared to the other baselines.
>
> [W3.] **Terseness of the write-up**  We agree with the reviewer that the Section 4 may be difficult to follow. Because a substantial part of our contribution is empirical, spanning many ablation studies on the importance of estimating joint versus input/conditional distributions, regression model variants, decision boundaries, as well as classification and segmentation tasks, it was not possible to include all results within the page limit. The main text presents the key takeaways, while the SI contains the full set of detailed experiments. If the paper is accepted, we will use the additional space to restructure and clarify the presentation of results in the **CRV** (camera-ready version).
>
> [Q1]: All the baselines are now cited in Section 4 of the main text and in the SI.
>
> [Q2]:  **On Posterior estimates** We thank the reviewer for the helpful suggestions regarding the posterior estimates. In the revised version, in addition to the results we have presented in the original version, we include additional results on posterior behavior within the training distribution, which can be used to identify the most challenging in-distribution samples. As shown in Figure 4, we provide the corresponding fits and percentile-based error bounds. An interesting observation is that our approach highlights the hardest samples to predict within the training set. A detailed explanation of these new findings is provided in the main text and in SI B.4.1.
>
> We hope that the revised version and our comments address all the concerns of the reviewer, particularly on the statistical significance of the results. We request the reviewer to kindly upgrade their assessment of our paper accordingly.

---

> > ### Comment · Reviewer_ToFP · 2025-11-28
> >
> > Thanks for the follow-up. You refer to the revised version in your response but I don't see any revisions to your PDF on OpenReview. Let me know if I am missing something,

---

> > > ### Author Response · Authors · 2025-11-28
> > > **Reply to the Reviewer**
> > >
> > > We thank the reviewer for the question. We have uploaded the revised pdf to the open review system and we are able to see our revised version. Could the reviewer please confirm if the version that they can access has a main text of 10 pages ? If the reviewer wishes, we can color/highlight our changes for quick review.

---

### Author Response · Authors · 2025-12-02
**Summary of the discussion for the benefit of the area chair**

Dear AC,

Given the unique circumstances prevailing for the rebuttal period, we are writing to you with a concise summary of our point of view regarding the rebuttal discussions regarding our paper. At the outset, we would like to thank you for stepping in for this new assignment at short notice.

In general, we had a constructive discussion with the reviewers and aimed to address all their points of criticism in our rebuttal and revised version (attached pdf). It is a pity that we could not complete the discussions. Nevertheless, we wanted to highlight the main points of criticism from the reviewers and our response below:

**Lack of comparison with Bayesian Approaches for OOD detection**: Two of the reviewers (ARrh and 2W7u) pointed out that we did not compare our method with Bayesian approaches. We conceded this point and as suggested by them, we compared our approach to two Bayesian approaches: MC-dropout and the very recent Rate-in algorithm. As we wrote to the reviewers, we have compared both these algorithms with our likelihood-based JLBC algorithm for the Wave Eqn. dataset and discussed the detailed algorithms and results in the main text as well as in SI B.1.7 and SI Table 4. As shown in this table, JLBC (ours) is significantly more accurate than the best performing Bayesian approach (Rate-in) with an ACC score of  0.87 (JLBC), compared to 0.76 (Rate-In). Moreover, JLBC is much faster to run at 0.02 secs, compared to the 0.15 secs of runtime for the Rate-In algorithm. Thus, we demonstrate that our proposed Diffusion-based approach is significantly superior to the Bayesian approaches. Needless to say, we will add these results into a CRV, if accepted, as they significantly strengthen the conclusions of our paper.


**On Theoretical Guarantees** Two of the reviewers (ARrh and 2W7u) also raised the point that the theoretical formulation of our certificate was inadequate. We agreed with the reviewers that the theoretical heuristics in Section 2 of the original version were not sufficiently clear. To address this and as we had stated in our reply to the reviewers, we provided a more formal derivation of the rationale behind using joint likelihood estimates as certificates, by establishing an upper bound on the error that explicitly incorporates the likelihood term. Under assumptions, we showed in SI A.1 that the (log-)error can be bounded by the joint likelihood with very high probability (see Eqn. 2 in the revised version of the paper (attached)). This bound makes explicit the connection between the joint likelihood and the pointwise errors, showing that the likelihood term naturally controls the behavior of the error and thus justifies its use as a certificate. We have provided the detailed derivation and critically examined the underlying assumptions in SI A.1. We hope that it provides a stronger theoretical foundation of our paper.

**Organization and References** All of the reviewers had raised certain minor points regarding the organization of the paper and adding some references. We have already addressed these comments in the rebuttal.

**General Summary** In conclusion, we found the points of criticism of the reviewers very constructive and have strived to address them during the rebuttal and in the revised version of the paper. One of the reviewers (27Wu) acknowledged the merit of the responses and raised our score to a *strong accept*. Due to the untimely termination of the rebuttal process, the other reviewers could not weigh in but we feel that they would be very positive. In particular, reviewer ARrh's points of concern were very similar to 27Wu and as 27Wu acknowledged that we had addressed these points, we expected that ARrh would share this judgement.

We hope that this summary is useful to the AC in making their final decision and are at your disposal during the rebuttal period for any further clarifications.

Yours sincerely'

Authors of *Towards a Certificate of Trust: Task-Aware OOD Detection for Scientific AI*

---

### Meta-Review · Area_Chair_U7aQ · 2026-01-07

**Summary:**

This paper proposes a task-aware out-of-distribution (OOD) detection method for scientific machine learning by estimating joint input–prediction likelihoods using score-based diffusion models. The resulting likelihood serves as a “certificate of trust” that strongly correlates with prediction error across regression, classification, and segmentation tasks in scientific domains. Extensive experiments show improved OOD detection over input-only likelihoods and several diffusion-based baselines.

**Reviewer Concerns:**

Overall, reviewers questioned (i) the strength and clarity of the theoretical justification linking joint likelihood to prediction error, (ii) comparisons to non-diffusion uncertainty/OOD baselines (e.g., Bayesian or uncertainty-based methods), (iii) computational cost and scalability of diffusion-based likelihood estimation, and (iv) presentation issues, including dense organization and reliance on supplementary material.

**Reviewer Scores:**

8,6,4

---

### Decision · Program_Chairs · 2026-01-26

Accept (Poster)